# TRANSLATING ROBOT SKILLS: LEARNING UNSUPERVISED SKILL CORRESPONDENCES ACROSS ROBOTS

## ABSTRACT

We explore how we can endow robots with the ability to learn correspondences between its own skills, and those of morphologically different robots in different domains, in an entirely unsupervised manner. Our insight and premise is that morphologically different robots use similar strategies (high-level skills sequences) to solve similar tasks. Based on this insight, we frame learning skill correspondences as a problem of matching distributions of sequences of skills across robots. We then present an unsupervised objective that encourages a learnt skill translation model to match these distributions across domains inspired by recent advances in unsupervised machine translation. Our approach is able to learn semantically meaningful correspondences between skills across 3 robot domain pairs despite being completely unsupervised. Further, the learnt correspondences enable the transfer of task strategies across robots and domains. Dynamic visualization of our results can be found here: https://sites.google.com/view/translatingrobotskills/home

## 1 INTRODUCTION

Humans have a remarkable ability to efficiently learn to perform tasks by watching others demonstrate similar tasks. For example, children quickly learn the skills needed to play a new sport by watching their parents perform skills such as kicking a ball. Notably, they are able to learn from these visual demonstrations despite significant differences between themselves and the demonstrator, including visual perspectives, environments, kinematic and dynamic properties, and morphologies. This ability may be attributed to two factors; firstly, humans have well-developed basic motor skills that we can execute with little effort. Second, we can recognize the sequence of skills (or the high-level strategy) the demonstrator uses, and understand a corresponding set of skills that we can execute ourselves (Meltzoff & Moore, 1977).

In this paper, we explore how we can endow robots with this ability - i.e., to adopt task strategies from morphologically different robot demonstrators, and then execute corresponding skills to solve similar tasks. Key to solving this problem is for the robot to identify how its owns skills correspond to those of the demonstrator, which is tremendously powerful. First, it allows the robot to adopt the demonstrator's strategies for solving various tasks. Second, by adopting and *adapting* these

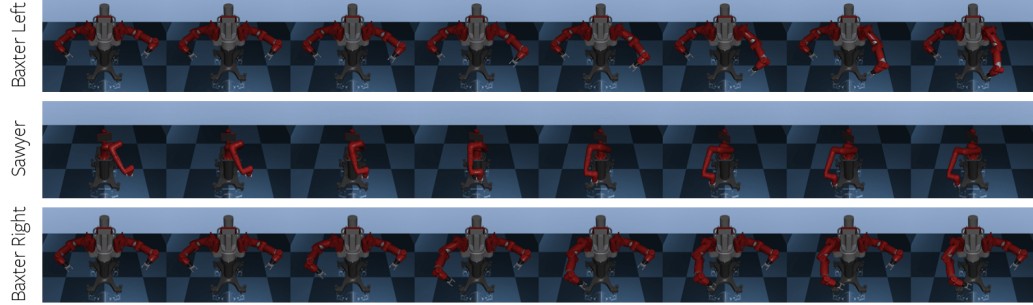

**Figure 1:** Sample skill correspondences learnt by our unsupervised approach, across three different morphological robots. We visualize a reaching skill on the Baxter left hand (top), translated to the Sawyer (middle) and the Baxter right hand (bottom) respectively.

strategies for itself, the robot can efficiently learn to solve a variety of tasks previously outside its repertoire. Finally, it allows us to understand the task strategies used by various robots in a unified manner, enabling robots to learn from data collected with a heterogeneous collection of tasks and robot morphologies. Skills provide a natural framework to facilitate such concise knowledge transfer across robots compared to low-level robot controls; skills inherently abstract away low-level details that may differ across the demonstrator and the learner, and instead focus on the commonalities between them, such as the task strategy. For example, skills such as reaching and placing on robot manipulators abstract away differences in morphologies or configuration, and are thus well grounded across robots, while there may not be obvious correspondence in their low-level actions.

How can one acquire correspondences between skills? Stated more formally, given morphologically different robots, a set of unlabelled demonstrations of each robot solving a variety of tasks, and a method for extracting skills from those demonstrations, how can one learn correspondences between the skills of the different robots? Learning such correspondences is straightforward when supervised pairs of skills or trajectories are available. However, collecting and annotating such data is time-consuming and tedious, and requires a high degree of human expertise, particularly at the scale necessary to successfully learn correspondences across a diverse set of skills. In contrast, we explore whether we can learn such skill correspondences in an *unsupervised* manner, i.e., without access to supervised skill or trajectory pairs. Learning *unsupervised* skill correspondences potentially allows for learning a broad scope of correspondences from the diverse unlabelled and unsegmented data newly available in robotics (Sharma et al., 2018; Mandlekar et al., 2018). Unfortunately, learning unsupervised correspondences is a difficult problem to solve - without supervision, it is difficult to guide learning towards the right correspondences. While this may be alleviated to some extent by incorporating unsupervised constraints (Zhu et al., 2017; Ganin et al., 2016; Zhou et al., 2019), there are many spurious correspondences that satisfy these constraints.

How can one then go about learning such skill correspondences in an unsupervised manner? We propose leveraging the two following insights to do so. Our first insight is that different morphological robots use similar task strategies (in terms of sequences of skills) to solve similar tasks; in other words, the sequences of skills executed by different robots to solve similar tasks ought to belong to similar *distributions*. We observe that this is generally true; for example, a robot pouring out a cup of tea would likely first reach for the kettle, grasp it, move it appropriately over the cup, and then begin to pour the tea, irrespective of its exact morphology. Our second insight is that learning skill correspondences without access to supervised data closely mirrors unsupervised machine translation (UMT), where the objective is to learn a translation between representations in different languages (such as between word embeddings across languages), without access to parallel data (Conneau et al., 2017; Lample et al., 2018). Inspired by these two insights, we derive an unsupervised objective to guide our learning towards meaningful skill correspondences.

We approach the problem of learning unsupervised skill correspondences by learning a translation model to map from the skill space on one "source" robot to the skill space on another "target" robot. We construct an unsupervised objective that encourages the translation model to respect our first insight - i.e., to preserve the sequences of skills observed across both the (translated source) and target robots. To do this, we take inspiration from our second insight, and specifically from Zhou et al. (2019), and construct explicit probability density models over the sequences of skills observed on the source and target robots, and then train the translation model to match these distributions.

We evaluate the ability of our approach to learn unsupervised skill correspondences across three different robots, the Sawyer robot, and the left and right hands of the Baxter robot. Our approach is able to learn semantically meaningful correspondences across each of the 3 pairs of robots, despite being completely unsupervised, as depicted in fig. 1. The learnt correspondences facilitate transferring task strategies across across domains, as we demonstrate on a set of downstream tasks. Our results are visualized at https://sites.google.com/view/translatingrobotskills/home.

## 2 RELATED WORK

**Skill Learning:** Eysenbach et al. (2019); Sharma et al. (2020) both address unsuperivsed skill learning from interaction data, by constructing information theoretic approaches. Fox et al. (2017); Krishnan et al. (2017); Shankar et al. (2020); Sharma et al. (2018); Shankar & Gupta (2020); Kipf et al. (2019); Gregor et al. (2019); Kim et al. (2019) instead learn skills or abstractions from unlabelled

demonstration data by performing latent variable inference. These frameworks all learn skills in the context of a single domain, while we seek to learn correspondences of skills *across* domains.

**Unsupervised Correspondence Learning:** Our problem bears a close resemblance with unsupervised machine translation (Conneau et al., 2017; Zhou et al., 2019), and unpaired image translation (Zhu et al., 2017; Park et al., 2020). Specifically, they all share the notion of learning correspondences across representations learnt from unpaired, "monolingual" data:

- Unsupervised Correspondence in Machine Translation: Conneau et al. (2017) leveraged domain-adversarial training (Ganin et al., 2016) to align word embeddings of two languages. Sennrich et al. (2016); Lample et al. (2018) use the idea of back-translation to constraint the learnt translation models across languages. Zhou et al. (2019) learn bilingual word embeddings by matching explicit density functions over the word embedding spaces across languages.
- Unsupervised Correspondence in Image Translation: The vision community has similarly explored the unpaired *image-to-image* translation setting (Zhu et al., 2017; Park et al., 2020), using cycle-consistency losses (Zhu et al., 2017) or contrastive losses (Park et al., 2020).
- Unsupervised Correspondence in Video: Bansal et al. (2018); Wang et al. (2019) extend Cycle-GAN (Zhu et al., 2017) to the video domain, by incorporating temporal consistency losses.

**Domain Transfer in Robotics and Graphics:** The robotics and graphics communities have taken interest in cross-domain transfer of policies in recent years:

- Policy Transfer with Paired Data: Gupta et al. (2017); Sermanet et al. learn morphology and viewpoint invariant feature spaces for policy transfer respectively, but require paired data to do so.
- Policy Transfer via Modularity: Hejna et al. (2020); Devin et al. (2017); Sharma et al. (2019) address morphological transfer by modularity in their policies. Hejna et al. (2020); Sharma et al. (2019) adopt modularity in a hierarchical sense, but leverage common grounding of subgoals across morphologies to perform transfer. We do not assume access to such common grounding.
- State based Policy Transfer: Liu et al. (2020); Schroecker & Isbell (2017); Ammar et al. (2015) address cross-morphology transfer by state based imitation learning.
- Motion Retargetting: The graphics community has addressed transferring behaviors across morphologically different characters (Hecker et al., 2008; Aberman et al., 2020; Villegas et al., 2018; Abdul-Massih et al., 2017), using carefully handcrafted kinematic models. These works transfer behaviors by imitating joint positions, and performing inverse kinematics to retrieve the full character state, which is often infeasible for widely different morphological characters.
- Unsupervised Action Correspondence: Zhang et al. (2021); Kim et al. (2020); Smith et al. (2020) address learning *low-level* state and action correspondences from unpaired, unaligned interaction and demonstration data respectively. While similar in spirit, our work argues that learning high-level skill correspondence instead is a more natural choice, as mentioned in the introduction.

**Applicability to learning skill correspondences:** While successful in their respective problem domains, existing approaches are not trivially applicable to our problem. The adversarial training used in most of these approaches is notoriously difficult to train, and is prone to the mode dropping problem (Li & Malik, 2019). They require strong constraints such as pixel-wise or joint-wise identity losses (Zhu et al., 2017; Zhang et al., 2021), or inherent similarity of the spaces to be aligned, such as word embeddings across languages (Conneau et al., 2017; Zhou et al., 2019). Learnt skill representations do not possess this property in general. Shortcomings notwithstanding, these approaches provide insight into how we may pursue learning unsupervised skill correspondences.

## 3 APPROACH

### 3.1 PRE-REQUISITES

We begin by first reviewing an important prerequisite - the skill learning pipeline of Shankar & Gupta (2020), which provides us a learnt representation of robotic skills given an unlabelled robot demonstration dataset. Their method first represents robotic skills as continuous latent variables $z$, and introduce a Temporal Variational Inference (TVI) to infer these skills or latent variables. TVI trains a variational encoder $q(z|\tau)$ that takes as input a robot trajectory $\tau = \{s_1, a_1, ...s_{n-1}, a_{n-1}, s_n\}$, and outputs a sequence of skill encodings $z = \{z_1, z_2, ...z_n\}$. Note that these skill encodings repeat over time for the duration of a given skill. TVI also trains a latent conditioned policy $\pi(a|s, z)$ that takes as input robot state $s$, and the chosen skill encoding $z$, and predicts the low-level action $a$ that the robot

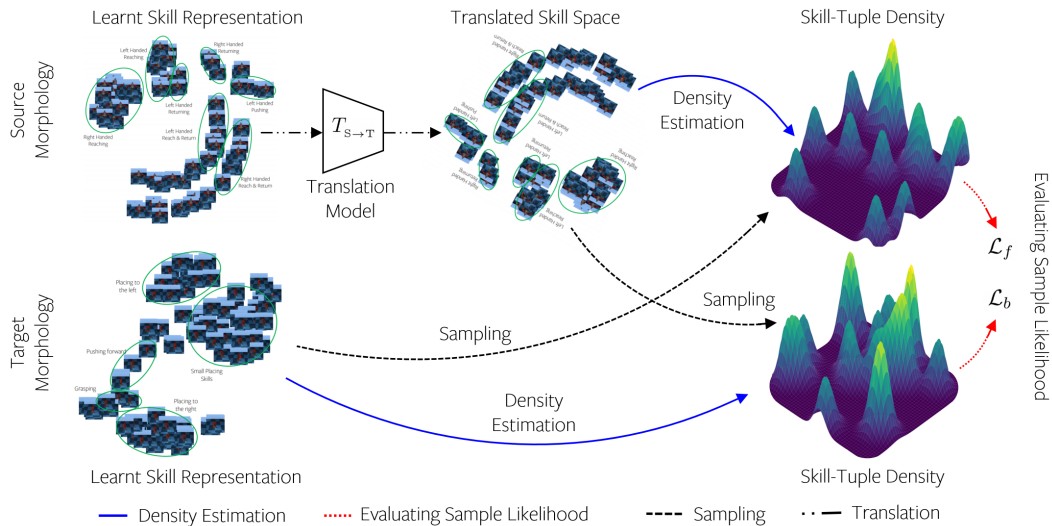

**Figure 2:** Overview of our approach. We translate the original learnt skill space to the target domain. We then construct explicit density estimates over the original target and translated source skill-tuple spaces. We train our translation model to maximize the likelihood of randomly sampled translated source and original target skill-tuples under these densities respectively, affording meaningful skill-correspondences across both robots.

should execute. We direct the reader to Shankar & Gupta (2020) for a more thorough description of their skill learning approach.

## 3.2 PROBLEM SETTING

Consider two robots with potentially differing morphologies and environments (or "domains"), represented as source and target domains, or $M_S$ and $M_T$ respectively. Associated with each domain is an unlabelled and unsegmented dataset of demonstrations of the robot performing various tasks represented as $D_S$ and $D_T$ respectively. $D_S$ and $D_T$ are collected independently, on an intersecting (but not identical) set of tasks. This is equivalent to the setting in machine translation without *parallel* data, with access only to *monolingual* corpora.

We also assume access to a skill-learning pipeline, that can take in such an unlabelled demonstration dataset $D$ of a robot $M$ performing various tasks, and learns a representation $\mathbb{Z}$ of skills on a given robot. We specifically use TVI (Shankar & Gupta, 2020), but we believe our approach is compatible with any unsupervised skill-learning approach that affords a continuous representation space. We train the skill-learning pipeline independently on each of the domain datasets $D_S$ and $D_T$. This affords us skill encoders $q_S$ and $q_T$, and latent conditioned policies $\pi_S$ and $\pi_T$ for each domain. We may then encode trajectories in both domains $\tau_S \in D_S$ and $\tau_T \in D_T$ into their constituent skills $\{z_S^t\}_{t=1}^{|\tau_S|}$ and $\{z_T^t\}_{t=1}^{|\tau_T|}$, via $q_S$ and $q_T$ respectively. We represent the space of skills learnt on the entire dataset in each of the two domains as $\mathbb{Z}_S$ and $\mathbb{Z}_T$ respectively.

Our goal is to learn semantically meaningful correspondences between skills in the source domain, $z_S \in \mathbb{Z}_S$, and those in the target domain, $z_T \in \mathbb{Z}_T$, that helps solve tasks in the target domain $M_T$. We specify these correspondences by learning a "translation" function $T_{S \to T} : \mathbb{Z}_S \to \mathbb{Z}_T$, that maps skills in the source domain $z_S$, to translated skills in the target domain $z_{S \to T}$. This mirrors the word translation models present in Conneau et al. (2017); Zhou et al. (2019). Learning meaningful skill-correspondences thus amounts to learning a good translation model $T$. We can also translate the entire source space $\mathbb{Z}_S$ to the target domain, to retrieve the translated source space, $\mathbb{Z}_{S \to T}$.

## 3.3 USING SEQUENTIAL INFORMATION OF SKILLS

Our first insight is that different morphological robots follow similar strategies to address similar tasks - in other words, sequences of skills executed by two different robots ought to belong to similar distributions. It is important to consider *sequences* of skills here rather than skills in isolation, since the ordering of skills gives us additional information to bias our learning towards the right correspondences. Constraining correspondences based on individual skills alone could lead to incorrect results. For example, learning that "reaching" skills in one domain correspond to "grasping" skills in the other domain is evidently incorrect, and is an error that sequential information could bias learning away from, since grasping skills are systematically executed *after* reaching skills.

However, processing entire sequences of skills is computationally infeasible for arbitrarily long trajectories. Instead, we draw inspiration from the simple yet powerful bigram models in the NLP community, and consider consecutive pairs, or *tuples* of skills. While less expressive than the full skill-sequences, these tuples retain enough information of the ordering of skills to guide the correspondence learning towards the right solution, and are far more computationally tractable. We thus construct a skill-tuple space, which represents the various transitions between skills observed in a given domain. Each consecutive pair of skills $(z^t, z^{t+1})$ in the original space $\mathbb{Z}$ of a given domain is represented as a single point $x^t$ in the skill-tuple space $\mathbb{X}$ of that domain. For initial and terminal skills $z^{t=0}$ and $z^{t=|\tau|}$, we pad these tuples, and treat the preceeding and succeeding skill encodings as appropriately dimensioned 0's respectively.

The skill-tuple spaces for source and target domains are represented as $\mathbb{X}_S$ and $\mathbb{X}_T$ respectively, and are implemented as a set of skill-tuples from each domain, i.e. $\mathbb{X}_S = \{x_{S,i}\}_{i=1}^{N_S}, \mathbb{X}_T = \{x_{T,i}\}_{i=1}^{N_T}$. The procedure to construct these spaces is presented in the sub-routine of algorithm 1. We also construct a translated skill-tuple space, $\mathbb{X}_{S\to T}$, that is simply the translation of all skills present in $\mathbb{X}_S$, i.e. $\mathbb{X}_{S\to T} = \{x_{S\to T,i}\}_{i=1}^{N_S}$. Translating a skill-tuple $x_S^t = (z_S^t, z_S^{t+1})$ from source to target is done by translating the individual skills $z_S^t$ and $z_S^{t+1}$ to the target domain, $z_{S\to T}^t$ and $z_{S\to T}^{t+1}$, and constructing the translated skill-tuple as $x_{S\to T}^t = (z_{S\to T}^t, z_{S\to T}^{t+1})$.

### 3.4 DISTRIBUTIONAL PERSPECTIVE ON LEARNING TRANSLATION FUNCTIONS

We would like our translation model to exhibit two properties - first, to translate source skills $z_S$ such that they belong to the distribution of target skills, and second, to capture all modes of skills that exist in the target domain. Dropping modes of skills limits the set of skills the target robot can use, reducing its capabilities. GANs (Goodfellow et al., 2014) and domain-adversarial approaches (Ganin et al., 2016) satify the first property since they optimize for how realistic the generated samples are (or how indistinguishable source and target domain features are (Ganin et al., 2016)), but often drop modes of the "real" data (Li & Malik, 2019).

Instead, we approach this problem from an *explicit* distribution perspective, and maintain explicit probability density estimates over the translated source and target skill-tuple spaces, $\mathbb{X}_{S\to T}$ and $\mathbb{X}_T$, $p(x_{S\to T})$ and $p(x_T)$ respectively. Following our insight that sequences of skills on different robots ought to belong to similar distributions, we would like these distributions $p(x_{S\to T})$ and $p(x_T)$ to match. We can optimize our translation model to enforce this, by maximizing the likelihood of translated skill-tuples $x_{S\to T}$ under the target skill-tuple distribution $p(x_T)$, and the likelihood of target skill-tuples $x_T$ under the translated skill-tuple distribution $p(x_{S\to T})$. This is similar in spirit to optimizing *both* the forward and reverse KL between two distributions. We formalize this objective and provide intuition for it below.

Consider constructing an explicit probability density function $p_T(x)$, over the target domain skill-tuple space $\mathbb{X}_T$. We follow Zhou et al. (2019)'s choice of representing $p_T(x)$ using a Gaussian Mixture Model (GMM), owing to its expressive power and efficiency in low-dimensional spaces, but note that a wide variety of explicit density estimators may be used here. Here, $p_T(x)$ is a GMM with $N_T$ Gaussian kernels, each centered at a target skill-tuple $x'_{T,i} \sim \mathbb{X}_T$:

$$p_T(x) = \sum_{i=1}^{N_T} p(x'_{T,i})\, p(x|x'_{T,i}) = \sum_{i=1}^{N_T} p(x|x'_{T,i}) \tag{1}$$

Zhou et al. (2019) use kernel weights $p(x'_{T,i})$ proportional to the frequency of the word the kernel represents. In our setting, each kernel is simply one of the skill-tuples in the *continuous* target skill space $\mathbb{X}_T$, motivating our choice of equal mixture weights over all kernels (i.e. skill-tuples), i.e. $p(x'_{T,i}) = 1/N_T$. The mixture components $p(x|x'_{T,i})$ are specified by a fixed variance Gaussian centered at $x'_{T,i}$, i.e. $p(x|x'_{T,i}) = \mathcal{N}(x|x'_{T,i}, \sigma^2)$, where $\sigma$ specifies the standard deviation of the Gaussian kernel. We can train a translation model to translate skills in such a way that the translated skill tuples $x_{S\to T}$ belong to the distribution of target skill-tuples $p_T(x)$, or that they are realistic with respect to the target distribution. We do this by maximizing the "forward" likelihood $\mathcal{L}_f$ of the translated skill-tuples $x_{S\to T}$, under the target density $p_T(x)$:

$$\mathcal{L}_f = \mathbb{E}_{x_S \sim p(x_S), x_{S\to T} \sim T_{S\to T}(\cdot|x_S)}\Big[\log p_T(x_{S\to T})\Big] \tag{2}$$

---

**Algorithm 1** Translating Robot Skills

---

**Input:** $D_{\mathrm{S}}, D_{\mathrm{T}}, q_{\mathrm{S}}, q_{\mathrm{T}}, \pi_{\mathrm{S}}, \pi_{\mathrm{T}}$
    ▷ Require demonstration datasets, trained skill encoders & decoders for source & target domains
**Output:** $T_{\mathrm{S}\to\mathrm{T}}(\,.\,|z_{\mathrm{S}})$                                                   ▷ Output translation model
 1: Initialize $T_{\mathrm{S}\to\mathrm{T}}(\,.\,|z_{\mathrm{S}})$                                      ▷ Initialize translation model
 2: $\mathbb{X}_{\mathrm{S}} \leftarrow$ BUILDSKILLTUPLESPACE(Source)       ▷ Construct source skill-tuple space
 3: $\mathbb{X}_{\mathrm{T}} \leftarrow$ BUILDSKILLTUPLESPACE(Target)        ▷ Construct target skill-tuple space
 4: $p_{\mathrm{T}}(x) \leftarrow \sum_{i=1}^{N_{\mathrm{T}}} p(\,.\,|x'_{\mathrm{T},i})$          ▷ Update target GMM density via eq. (1)
 5: **for** $i \in [1, 2, ..., N_{\mathrm{iterations}}]$ **do**
 6:     $\mathbb{X}_{\mathrm{S}\to\mathrm{T}} \leftarrow T_{\mathrm{S}\to\mathrm{T}}(\,.\,|\mathbb{X}_{\mathrm{S}})$         ▷ Update translated-source skill-tuple space
 7:     $p_{\mathrm{S}\to\mathrm{T}}(x) \leftarrow \sum_{i=1}^{N_{\mathrm{S}}} p(\,.\,|x'_{\mathrm{S}\to\mathrm{T},i})$   ▷ Update translated-source GMM density via eq. (3)
 8:     $x_{\mathrm{S}} \sim \mathbb{X}_{\mathrm{S}}, x_{\mathrm{T}} \sim \mathbb{X}_{\mathrm{T}}$        ▷ Get batch of source and target skill-tuples
 9:     $x_{\mathrm{S}\to\mathrm{T}} \sim T_{\mathrm{S}\to\mathrm{T}}(\,.\,|x_{\mathrm{S}})$        ▷ Translate source skill-tuple to target domain
10:     $\mathcal{L}_f \leftarrow \log p_{\mathrm{T}}(x_{\mathrm{S}\to\mathrm{T}})$             ▷ Evaluate forward objective
11:     $\mathcal{L}_b \leftarrow \log p_{\mathrm{S}\to\mathrm{T}}(x_{\mathrm{T}})$             ▷ Evaluate backward objective
12:     $\mathcal{L} \leftarrow \mathcal{L}_f + \mathcal{L}_b$                   ▷ Evaluate full objective
13:     Update $T_{\mathrm{S}\to\mathrm{T}}$ via $\nabla \mathcal{L}$        ▷ Update translation model with gradient ascent

---

**Sub-routine**: Build Skill-Tuple Space

---

**Input:** Domain $M$
**Output:** Skill-Tuple space $\mathbb{X}_M$
14: **function** BUILDSKILLTUPLESPACE(Domain $M$)
15:     $\mathbb{X}_M \leftarrow \{\}$                              ▷ Initialize empty skill-tuple space
16:     **for** $i \in [1, 2, ..., N_{\mathrm{M}}]$ **do**
17:         $\tau_{\mathrm{M},i} \sim D_{\mathrm{M}}$       ▷ Retrieve batch of trajectories from domain dataset
18:         $\{z_{\mathrm{M}}^t\}_{t=1}^{|\tau|} \sim q_{\mathrm{M}}(\,.\,|\tau_{\mathrm{M},i})$ ▷ Encode trajectories as sequence of skills via domain encoder
19:         $\{x_{\mathrm{M}}^t\}_{t=1}^{|\tau|-1} \leftarrow \{(z_{\mathrm{M}}^t, z_{\mathrm{M}}^{t+1})\}_{t=1}^{|\tau|-1}$            ▷ Assemble skill tuples
20:         $\mathbb{X}_M \leftarrow \mathbb{X}_M \cup \{x_{\mathrm{M}}^t\}_{t=1}^{|\tau|-1}$         ▷ Add skill-tuples to skill-tuple space
21:     **return** $\mathbb{X}_M$

---

To encourage the second mode-covering property in the translation model, we can also optimize the likelihood of the *target* skill-tuples under the *translated source* skill-tuple space. To do so, we first construct an explicit probability density function $p_{\mathrm{S}\to\mathrm{T}}(x)$ over the translated source skill-tuple space, $\mathbb{X}_{\mathrm{S}\to\mathrm{T}}$, that is equivalent to $p_{\mathrm{T}}(x)$. $p_{\mathrm{S}\to\mathrm{T}}(x)$ is also similarly represented as a GMM, with $N_{\mathrm{S}\to\mathrm{T}}$ equally weighted Gaussian kernels, each centered at a *translated* skill-tuple $x'_{\mathrm{S}\to\mathrm{T},i} \sim \mathbb{X}_{\mathrm{S}\to\mathrm{T}}$:

$$p_{\mathrm{S}\to\mathrm{T}}(x) = \sum_{i=1}^{N_{\mathrm{S}\to\mathrm{T}}} p(x'_{\mathrm{S}\to\mathrm{T},i})\, p(x|x'_{\mathrm{S}\to\mathrm{T},i}) = \sum_{i=1}^{N_{\mathrm{S}\to\mathrm{T}}} p(x|x'_{\mathrm{S}\to\mathrm{T},i}) \tag{3}$$

As before, the conditional $p(x|x'_{\mathrm{S}\to\mathrm{T},i}) = \mathcal{N}(x|x'_{\mathrm{S}\to\mathrm{T},i}, \sigma^2)$. We may then optimize the "backward" likelihood $\mathcal{L}_b$ of a set of target skill-tuples $x_{\mathrm{T}}$ under $p_{\mathrm{S}\to\mathrm{T}}(x)$:

$$\mathcal{L}_b = \mathbb{E}_{x_{\mathrm{T}} \sim p(x_{\mathrm{T}})}\Big[\log p_{\mathrm{S}\to\mathrm{T}}(x_{\mathrm{T}})\Big] \tag{4}$$

By combining these two objectives we may construct our final objective $\mathcal{L}$ for training the translation model, by simply combining these two objectives, $\mathcal{L} = \mathcal{L}_f + \mathcal{L}_b$:

$$\mathcal{L} = \mathbb{E}_{x_{\mathrm{S}} \sim p(x_{\mathrm{S}}), x_{\mathrm{S}\to\mathrm{T}} \sim T_{\mathrm{S}\to\mathrm{T}}(.|x_{\mathrm{S}})}\Big[\log p_{\mathrm{T}}(x_{\mathrm{S}\to\mathrm{T}})\Big] + \mathbb{E}_{x_{\mathrm{T}} \sim p(x_{\mathrm{T}})}\Big[\log p_{\mathrm{S}\to\mathrm{T}}(x_{\mathrm{T}})\Big] \tag{5}$$

**Parsing the objective:** In contrast with adversarial training methods, which require alternating training phases and stability tricks such as gradient penalties, our objective amounts to simple maximum likelihood (albeit in two directions). It is hence simpler, more stable, and quicker to converge. Intuitively, $\mathcal{L}_f$ captures how "realistic" each translated skill-tuple looks with respect to the target skill-tuple distribution, while $\mathcal{L}_b$ captures how well the translated skills cover the modes of the target skill-tuple space. We present our full approach for learning unsupervised skill correspondences in algorithm algorithm 1, and a pictorial representation of it in fig. 2.

# 4 EXPERIMENTS

**Robots:** We evaluate our approach's ability to learn skills correspondences across the following robots - the Sawyer robot, the Baxter left hand, and the Baxter right hand. Our choices of robots are dictated by the availability of demonstration datasets for these respective robots. We consider 3 of the possible 6 unique domain pairs - Baxter Right-Hand to Baxter Left-Hand (Bax-R to Bax-L), Baxter Left-Hand to Sawyer (Bax-L to Saw), Baxter Right-Hand to Sawyer (Bax-R to Saw).

**Datasets:** For the Sawyer robot, we make use of the Roboturk dataset (Mandlekar et al., 2018), which consists of roughly 2000 demonstrations across 8 different tasks. For the Baxter robot, we utilize the MIME dataset (Sharma et al., 2018). We partition the MIME dataset into two disjoint sets with solely left-handed and right-handed trajectories respectively. Each single-handed dataset also roughly contains 2000 demonstrations across 16 tasks. We treat each single-handed dataset as the respective dataset for the Baxter left-hand robot and the Baxter right-hand robot.

**Skill Representation:** We learn skill-representations for each of the 3 robots independently from their respective datasets, using TVI (Shankar & Gupta, 2020). We use a 16-dimensional skill-representation space for each robot, that is learnt from robot joint-states and joint velocities. We then freeze these learnt skill representations for all of our experiments. We follow the preprocessing steps and training parameters specified by (Shankar & Gupta, 2020). We also manually annotate 50 skills in each domain with one of 6 semantic labels of what type of skills they were. We emphasize that our approach is not trained with these labels, but rather is only evaluated against them.

## 4.1 LEARNING MEANINGFUL SKILL CORRESPONDENCES

The first question we would like to answer is - *"Can our unsupervised method learn meaningful skill-correspondences between skill spaces?"* We first present qualitative results towards this.

**Translating Individual Skills and Entire Trajectories:** We present two sets of visualizations; individual skills and their translations, as well as entire trajectories (or sequences of skills) and their translations. We first visualize a set of source domain trajectories $\tau_{\mathrm{S}}$, obtained by rolling out their skill encodings $\{z_{\mathrm{S}}^t\}_{t=1}^{|\tau|}$ via the source domain policy $\pi_{\mathrm{S}}$. We then translate these sequences of skill encodings to the target domain, i.e. $\{z_{\mathrm{S} \to \mathrm{T}}^t\}_{t=1}^{|\tau|}$, and visualize rollouts of the *target* policy $\pi_{\mathrm{T}}$ conditioned on these translated skills. We present visualizations of individual skills and their translations, as well as visualizations of entire trajectories and their translations in fig. 1 and at https://sites.google.com/view/translatingrobotskills/home.

**Analysing Qualitative Skill Translations:** Despite being learned in a purely unsupervised manner, we observe our translation model is able to learn good semantically meaningful *coarse* skill correspondences between semantic clusters of skills that emerge in the original spaces, across all 3 domain pairs. For example, from fig. 1, we see source domain reaching skills correspond to target domain reaching skills across all 3 domain pairs, as do placing and pushing / sliding skills to the left or to the right, and returning skills to their rest configuration. However, our model is unable to guarantee learning perfect *finer* correspondences (such as between twisting or grasping skills), or between variations of skills that belong to the same semantic cluster in the original skill spaces (such as reaching with different arm shapes, as in fig. 1). We emphasize that our method is unsupervised, and instead exploits similar contexts of skills across domains to learn correspondences; these results are understandable and expected as a result. Finer skills and intra-cluster variations of such skills are often executed in similar contexts, such as after reaching or before placing skills, making them difficult to disambiguate. We note that despite this, our model often does translate finer skills (such as twisting and grasping) correctly, suggesting its potential for improvement via techniques such as iterative refinement (Conneau et al., 2017), or via additional inductive biases.

Using the learnt correspondences, we are able to transfer the source domain trajectory's overall structure, and sequence of skills executed, remarkably well to the target domain. Our model does so despite variations in the precise shape of the robot arms, or start and end positions of these respective skills. This provides further evidence that the learnt correspondences are indeed semantically meaningful, and additionally suggests these learnt correspondences would be useful in transferring *task strategies* across domains. We believe this is a powerful result, especially so considering approach does so in a completely unsupervised manner.

**Table 1:** Evaluating Skill-Tuple Distribution Matching via our approach against various baselines, across all domain pairs. Baselines are adapted from (1): Ganin et al. (2016), (2): Zhu et al. (2017), (3): Liu et al. (2020). Lower is better for all metrics except the label accuracy.

| Domain Pair | Approach: | $\mathcal{L}_f$ | $\mathcal{L}_b$ | Chamfer Distance | Cycle Error | Label Accuracy | Supervised Z Error |
|---|---|---|---|---|---|---|---|
| Bax-R to Bax-L | DomAd [1] | 38.1 | 43.4 | 20.1 | 17.4 | 10.3% | 15.2 |
| | Cycle GAN [2] | 32.1 | 39.8 | 17.0 | **9.2** | 14.2% | 14.9 |
| | State Im [3] | 45.3 | 49.1 | 31.0 | 24.8 | 32.8% | 12.3 |
| | **Ours** | **14.3** | **20.1** | **8.2** | 10.2 | **73.8%** | **8.1** |
| Bax-L to Saw | DomAd [1] | 50.9 | 60.1 | 24.2 | 21.3 | 9.8% | 14.9 |
| | Cycle GAN [2] | 38.5 | 53.7 | 17.1 | **8.8** | 12.0% | 15.4 |
| | State Im [3] | 66.5 | 73.1 | 38.9 | 22.4 | 19.3 % | 36.8 |
| | **Ours** | **16.8** | **33.9** | **7.8** | 9.0 | **68.7%** | **9.0** |
| Bax-R to Saw | DomAd [1] | 54.8 | 80.2 | 21.2 | 24.8 | 11.1% | 19.1 |
| | Cycle GAN [2] | 47.3 | 78.5 | 16.7 | 10.3 | 16.8% | 20.1 |
| | State Im [3] | 72.5 | 85.4 | 33.9 | 27.7 | 20.1% | 23.0 |
| | **Ours** | **15.4** | **36.5** | **7.2** | **10.1** | **64.6%** | **12.1** |

**Quantitative Evaluation of Skill-Tuple Distribution Matching:** To quantitatively measure how well our approach can match the distributions of skill-tuples across robots, we evaluate the following unsupervised metrics. We first evaluate the forward and backward GMM densities $\mathcal{L}_f$ and $\mathcal{L}_b$. We also compute the *Chamfer distance* (Fan et al., 2017) between the skill-tuple spaces, which represents the nearest neighbor distances across two point sets. Together, these three metrics specify how close the skill-tuple distributions across domains are. We also evaluate the following *supervised* metrics, using the manually annotated semantic labels associated with 50 skills in each domain. We reiterate these labels are only used for evaluation purposes, and are unseen at train time. First, we evaluate how accurately the learnt correspondences capture semantics of skills across domains, by measuring how well the semantic labels associated with a set of skills in the source domain match with those in the target domain upon translation, reported as the *label accuracy* in table 1. We also evaluate a *supervised Z error* - i.e. the average distance in latent space between the translated source domain skills, and the nearest target domain skills with the same semantic label.

We compare our approach against the following alignment baselines, and present the results in table 1. Our first baseline is *Domain-adversarial training* (Ganin et al., 2016), where we match skill-tuple distributions by training the translation model to fool a discriminator network trained to identify domains given skill-tuples. We also compare against *Cycle-GAN* (Zhu et al., 2017), where we train two translation models between skills from source and target domains to be cycle-consistent with one another, while also optimizing a domain-adversarial style objective (Ganin et al., 2016). Finally, we also compare against a *State-based Imitation* approach (Liu et al., 2020), where we transfer trajectories across domains by copying end-effector states across robots, using inverse kinematics to retrieve the closest feasible joint state.

**Analysis of Qualitative Results of Skill-Tuple Distribution Matching:** From table 1, we observe that our approach is notable able to learn correspondences that achieves a much higher semantic label transfer accuracy, consistent with our approach learning good *coarse* correspondences. The baseline approaches in contrast, are only able to achieve random-level label transfer accuracy, indicating their inability to learn correct correspondences. For example, the domain adversarial baseline ends up mapping several different types of skills in the target domain to single modes of skills in the target domain, across all 3 domain pairs. The Cycle-GAN baseline somewhat mitigates this approach, but takes a shortcut in the learning and simply places a single source skill nearby every mode of target skill. In contrast, our approach explicitly optimizes for distribution matching, and achieves significantly better unsupervised distribution matching metrics than the implicit baseline approaches. The state based imitation approach requires careful engineering to match end-effector states, and even so often fails to find feasible joint configurations while imitating trajectories.

**Table 2:** Evaluating Task-Strategy Transfer: Evaluating average rewards over 10 episodes obtained via our approach with and without finetuning against a hierarchical RL baseline, adapted from Kulkarni et al. (2016), across 3 tasks. Source domain results are shared across approaches, and are trained via Kulkarni et al. (2016).

| Domain Pair | Approach: | Downstream Task | | | | | |
|---|---|---|---|---|---|---|---|
| | | Reach (Source) | Reach (Target) | Push (Source) | Push (Target) | Slide (Source) | Slide (Target) |
| Bax-R to Bax-L | HRL [1] | | 12.2 | | 12.6 | | 10.7 |
| | Ours (Zero Shot) | 16.4 | 19.2 | 30.1 | 17.2 | 21.3 | 13.4 |
| | Ours (Fine-tune) | | 20.1 | | 19.1 | | 18.9 |
| Bax-L to Saw | HRL [1] | | 13.2 | | 11.3 | | 17.2 |
| | Ours (Zero Shot) | 14.6 | 18.6 | 33.4 | 34.5 | 20.4 | 24.6 |
| | Ours (Fine-tune) | | 19.1 | | 37.1 | | 38.1 |
| Bax-R to Saw | HRL [1] | | 15.2 | | 12.6 | | 10.7 |
| | Ours (Zero Shot) | 16.4 | 20.8 | 30.1 | 28.5 | 21.3 | 29.1 |
| | Ours (Fine-tune) | | 29.7 | | 32.5 | | 30.5 |

## 4.2 TRANSFERRING TASK STRATEGIES ACROSS DOMAINS

If our learnt correspondences capture our first insight, i.e. that different robots use similar skill sequences to address similar tasks, it follows that we can transfer task strategies across these robots by translating the skill sequence that specifies the task strategy. We evaluate how well the learnt correspondences help transfer task strategies across robots as follows.

**Task Strategy Transfer Setup:** We consider a set of tasks adapted from Mandlekar et al. (2018), described in the appendix. We then train high-level policies (Kulkarni et al., 2016) in the source domain to predict sequences of skills (i.e., the task strategy) to execute on each of these tasks. We then evaluate how well the *translation* of this task strategy performs on the same tasks in the *target domain*, both without fine-tuning (i.e., in a zero-shot manner), and after fine-tuning for 50 episodes. We evaluate this transfer against a hierarchical RL baseline, i.e. a high-level policy trained in the target domain over 500 episodes, and report our results in table 2. We also provide visualizations of the rollouts from the original and translated strategies in https://sites.google.com/view/translatingrobotskills/home.

**Analysing results on task strategy transfer:** We observe that the task strategies translated by our method are able to achieve appreciable task performance across all domain pairs and tasks, even without fine-tuning these strategies. Without fine-tuning, we see that the translated task strategies follow a semantically reasonable sequence of skills given the target task, achieving an appreciable task reward, but often reach slightly differing goal states than desired. Given our translation model only observes skill encodings, and no additional state information, this is expected. Upon fine-tuning these translated task strategies in the target domain, we observe a consistent increase in task performance, since fine-tuning allows for picking slightly different (but semantically similar) skills that reach more appropriate goal locations, etc. In contrast, hierarchical RL on the target domain is often slow to converge, and must randomly explore appropriate skill sequences. Even with fine-tuning, our approach needs roughly 10 times less training episodes to superior performance than that of the hierarchical baseline. These results suggest that our approach does indeed learn correspondences that facilitate transferring task strategies across robots.

## 5 CONCLUSION

We introduce a purely unsupervised approach to learn skill correspondences across different morphological robots from different domains. Our approach learns semantically meaningful correspondences across 3 robot domain pairs, and helps transfer task-strategies across domains, without the need for any fine-tuning, despite being completely unsupervised. We believe our approach could enable learning of correspondences across more general temporal abstractions, such as between skills and language instructions, or between skills and human video demonstrations. We hope this work helps enable robots learn from data from heterogenous robots in different domains, and helps further the state of robot learning from demonstration.

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

# A   APPENDIX

## A.1   IMPLEMENTATION DETAILS

### A.1.1   NETWORK ARCHITECTURES

We parameterize the various functions involved, namely the source and target domain encoders and low-level policies $q$ and $\pi$ respectively, and the learnt translation model $T_{\text{S}\to\text{T}}$ as neural networks, with following specific architectures.

1. Variational encoders $q$: In each domain, the variational encoder $q$ is parameterized as a 4 layer LSTM network, with a hidden size of 48 and ReLu activation layers. We further use an input layer from the appropriate input dimensions prior to the LSTM, and two output layers to predict a Gaussian mean and variance of the skill encodings predicted by the variational encoders.

2. Low-level policies $\pi$: In each domain, the low level policies $\pi$ are parameterized as 3 layer LSTM networks, with hidden sizes of 48 and ReLu activation layers. As above, we use appropriately sized input layers prior to the LSTM, and two output layers after the LSTM to predict a Gaussian mean and variance of the low-level actions output by the policy.

3. Translation model $T_{\text{S}\to\text{T}}$: The translation model is parameterized as a simple 4 layer MLP, with a hidden size of 48 units, and ReLu activation layers. We similarly use two output layers to predict a Gaussian mean and variance of the *translated* skill encodings $z_{\text{S}\to\text{T}}$.

4. Latent Skill Encoding Dimension: We use a skill encoding dimension of 16 across all domains / robots.

5. RL High-level policy: While training RL, we parameterize the high-level policy as a 4 layer MLP, with a hidden size of 48 units, and ReLu activation layers. As above, we use two output layers to predict a Gaussian mean and variance of the predicted skill encodings, in whichever domain we train the policy in.

### A.1.2   TRAINING DETAILS

While training the variational encoders and low-level policies, we follow the training procedure specified by Shankar & Gupta (2020). We then freeze the variational encoders and low-level policies obtained, before training our translation models. While training our translation models, we simply optimize $\mathcal{L} = \mathcal{L}_f + \mathcal{L}_b$ using the Adam optimizer (Kingma & Ba, 2014), implemented in Pytorch. We use the default parameters of Adam, i.e. a learning rate of $10^{-4}$.

### A.1.3   RL TRAINING DETAILS

While training the downstream RL, we implement a hierarchical version of Proximal Policy Optimization (Schulman et al., 2017), by adapting Achiam (2018). This mirrors the hierarchical RL setup of Kulkarni et al. (2016). We fix the low-level policies provided to the algorithm, and only train the high-level policies that predict the skill encodings fed into the low-level policies. The hierarchical RL baseline Kulkarni et al. (2016) is trained for 500 episodes, which we found to be sufficient for the algorithm's performance to saturate. The fine-tuning approach was allowed a budget of 50 episodes to adapt the translated task strategy. The results were evaluated against a fixed random seed of 0.

### A.1.4   RL TASKS

We train on the following set of tasks adapted from Mandlekar et al. (2018). In particular, we create instances of these tasks on each of the 3 different robots. The only differences between the environments across different robots comes in the form of different initialization states for the objects concerned,

1. Reach: The robot must execute a sequence of skills to reach a block placed on the table. The robot gets a shaped reward based on the distance from the block, and a binary reward upon reaching within a threshold distance of the block.

2. Push: The robot must execute a sequence of skills to push a red block and a green block together. The reward received has a shaped component based on the distances of the end effector from the

    red block, as well as a binary reward based on whether the blocks are within a threshold distance of one another.

3. Slide: The robot must execute a sequence of skills to push a red block within a threshold distance of a green block placed farther away. The reward received is based on the distance of the end effector from the red block, as well as a binary reward based on whether the blocks are within a threshold distance of one another.

### A.1.5 HYPERPARAMETERS

We provide a list of the various hyperparameters use during training of the translation model itself, and the downstream RL training.

1. Number of GMM Kernels $N_S$ and $N_T$: For each of the forward and backward GMMs, we use $500$ kernels to parameterize the GMM.

2. GMM Kernel variance $\sigma^2$: For both forward and backward GMMs, we use a Gaussian Kernel variance of $0.5$.

3. Relative weighting of forward and backward losses: We weight the forward and backward losses equally during training.

4. Learning Rate: For training our translation model, we use the Adam optimizer with a learning rate of $10^{-4}$.

5. Batch Size: For our training, we use a batch size of $32$.

6. Number of iterations: We train our translation models over $8000$ epochs for each domain pair.

7. Random Seed: We set the random seed for our training to $0$ manually.

8. Epsilon Noise: We add in epsilon noise to our training during sampling from the learnt networks. Here, we use an initial epsilon value of $0.3$, and decay the epsilon value to $0.1$ over $200$ epochs.

While training downstream RL, we adopt the following hyperparameters:

1. Random Seed: When training downstream RL, we report results across 3 different seed values, $0, 1, 2$.

2. Epsilon Noise: We follow an epsilon-greedy exploration process during RL training, and use an initial epsilon value of $0.7$, and decay the epsilon value to $0.3$ over $200$ epochs.

3. PPO Parameters: We follow the default PPO parameters used in Achiam (2018).

### A.2 ASSUMPTIONS OF LEARNING

1. *Assumptions of datasets:* As stated in our main paper, our choice of robots is dictated by the availability of demonstration datasets on a particular robot. Further, the demonstration datasets ideally also exhibit the following traits. The demonstration datasets need to be *directed.*, i.e. they need to contain demonstrations of the robots solving a set of tasks, rather than containing random play data in an environment. While it is possible to learn skills from such play data, learning correspondences of skills from such play data is difficult. This is because our approach relies on directed sequences of skills to learn correspondences. Random play data often contains random sequences of skills, and so observe skills in contexts that they do not originally occur. This misleads our approach, which seeks to exploit context to learn correspondences.

2. *Assumptions about learnt skill spaces:* As stated in the analysis of our methods translations of skills, our model is able to learn good *coarse* correspondences, between clusters of skills in the original skill spaces. One trait of the original skill spaces that allows for this is the disentangled representation of the skills afforded by the skill-learning pipeline. Since similar semantic skills that occur in similar contexts are placed in similar clusters in the original skill space, this allows our approach to learn correct correpsondences between clusters of skills.

### A.3 ABLATION STUDIES

To accurately assess our contribution, we would also like to quantify how much each of the following components in our approach contribute to successfully learning correspondences:

1. The forward objective, $\mathcal{L}_f$.
2. The backward objective, $\mathcal{L}_b$.
3. The use of sequential information, i.e. matching skill-tuple distributions as opposed to matching *skill* distributions.
4. Learning a translation model across *frozen* skill spaces, as opposed to directly optimizing the skill representation itself.

We do so by removing each of these components individually, while keeping the rest of the method as is, and train our approach on all 3 domain pairs. We observe the following.

1. Without forward objective $\mathcal{L}_f$, i.e., only optimizing the backward objective $\mathcal{L}_b$, we observe the translation model is able to cover each of the modes of the target skill-tuple by translated source skills. However, since there is no objective that encourages how *realistic* the translated source skills are, we observe that there are several modes of translated skills that are *not* observed in the target skills space. This results in spurious correspondences. For example, this translation model learns to map reaching skills to $z$'s that are decoded into random jerky motions on the target robot, that are out of distribution for the target robot decoder.

2. Without backward objective $\mathcal{L}_b$: While only optimizing the *forward* objective $\mathcal{L}_f$, we observe that each of the translated skills appear very realistic with respect to the target domain, i.e. each of the translated skills look like a skill in the target domain. However, these models end up suffering from the same issue as the GAN based approaches or the domain adversarial style training, where several modes of the target skill space go unrepresented by target skills. This limits the ability of the learnt correspondences to effectively transfer strategies or represent source motions, since many of the target robot skills are never chosen by the translation model. Together, these ablations indicate the importance of both directions of our approach in learning successful correspondences.

3. Without sequential information: We can also optimize our objective defined in terms of skills themselves, rather than skill-tuples. While matching skill distributions, as opposed to skill-tuple distributions, we observe that the densities of skill distributions can be matched well by our objective. However, without access to sequential information, the model learns correspondences that are often wrong. For example, it learns to map reaching skills on the Baxter right hand to returning skills on the Baxter left hand, and vice versa. Skills in different contexts are often mapped to each other, since there is no contrary sequential information that the model has access to to suggest otherwise. This suggests that as described in our main paper, learning with access to sequential information and matching skill *tuple* distributions is also key to the success of our approach.

4. By directly optimizing the skill-representations, rather than training a translation model with fixed skill spaces: Instead of freezing the source and target skill spaces, we can also directly optimize the skill representations based on our objective. One may also think of this as maintaining a translation model that is simply an identity function. When allowing either or both of the source or target skill spaces to be finetuned, we must also optimize the original reconstruction style objective that is used in Shankar & Gupta (2020). Despite this, we observe a collapse of the skill space into a unimodal distribution, that is both unable to reconstruct skills in either domain, and be mapped across domains. This suggests freezing the representations and training a separate translation model is key to our approach.

