# OpenReview forum: "Translating Robot Skills: Learning Unsupervised Skill Correspondences Across Robots"
_ICLR.cc/2022/Conference — ICLR 2022 Submitted_

### Official Review · Reviewer_rKBg · 2021-11-01

**Correctness:** 3
**Technical Novelty And Significance:** 3
**Empirical Novelty And Significance:** 3
**Recommendation:** 6
**Confidence:** 4

**Main Review:**

The paper is written clearly and is well structured. The contribution is presented in a clear way, the technical formulation is sound and claims are supported by experimental results.
The visual representations (diagrams, figures and videos) are clear and help understand the contribution.
Implementation details are provided and comparisons with alternative methods allow to appreciate the benefits of the proposed method.
It could be argued that the reach skill is not complex enough to appreciate the accuracy of the proposed method in translating from one domain to another. This is partly captured by the results in table 2. Can you discuss these in more detail? In particular, can you explain the cases where the reward in the target domain is greater than that in the source domain?
How do you expect your method to transfer to real robots?


**Summary Of The Paper:**

This paper frames learning skill correspondences as a problem of matching distributions of sequences of skills across robots, and presents an unsupervised objective that is able to learn semantically meaningful correspondences between skills across diverse robot domains.


**Summary Of The Review:**

The paper presents a method that is well formulated, sound and well motivated. The experimental results are convincing but require some further explanation. Details of implementation and experimental results are provided, so that it could be possible to reproduce the results.

---

> ### Author Response · Authors · 2021-11-16
> **Response to Reviewer rKBg**
>
> We would like to thank the reviewer for their review and their feedback. We thank the reviewer for appreciating our paper being clear and well structured, and appreciating our formulation and the results and figures that support it.
>
> We address the specific concerns raised by the reviewer below -
>
> 1. On the evaluation of translations from our proposed method -
> We evaluate our approach’s ability to translate skills across the various skills that we observe in the original skill spaces on each of the robots. This includes a variety of skills, such as reaching, pushing, placing, wiping, twisting, etc. We note that our results in Table 1 are evaluated across all of these skills (and other skills that occur in the original skill spaces in each domain). As the reviewer notes, our visual representations also contain visualizations of various different skills and their respective translation, including reaching, pushing, placing, wiping, twisting, etc. Further, we depict various sequences of skills and their respective translations across domains. The other tasks in Table 2 also first reaching to an appropriate position, and subsequently executing pushing or sliding skills. We hope these results together convince the reviewer of the ability of our approach to come up with semantically meaningful skill correspondences despite being unsupervised.
>
> 2. On the reward magnitudes in source and target domains -
> We note that reward values are meant to be indicative of relative performance of various approaches within a given domain, rather than comparing performance across source and target domains. This is because of the following factors. The initial object states and goal states observed in the RL evaluation episodes are sampled from a preset distribution of initial states. By fixing the random seeds, we ensure they are the same across various approaches in a given domain for a given task. However, they may be different across domains, since the mean parameters of these distributions vary across domains. For example - across the left and right hand of the Baxter, the initial state distribution is set to be centered at a certain distance from the respective arm. The initial object position is sampled from this distribution, leading to different object states across domains. This leads to different reward values across source and target domains, that may be higher in the target domain than the source domain. We reiterate that the key observation here is the relative performance of our approach (with and without finetuning) over the baseline RL approach.
>
>
> 3. On how we expect our method to transfer to real robots - We first note that the key factor in understanding how well our method performs on real robot is the reliability of execution of skills on the real robot. Executing such skills on a real robot typically results in marginally more noisy trajectories than when executed on a simulated robot, owing to minor inaccuracies in the execution of control commands on the real robot. The translated skills (that are similar to skills from the target domain) would hence suffer from being slightly more noisy than their simulated counterparts. However, the skills afforded to us by Shankar & Gupta (2020) possess some amount of inherent spatial invariance. This suggests that execution noise observed on real robots would not affect the resultant trajectories by a large extent. We hope this convinces the reviewer our method would perform well on translating skills across different real robots.
>
> We hope these clarifications allay the reviewer's concerns with respect to the experimental results presented, and help them appreciate the strengths of our paper more unequivocally. We shall add these clarifications to our paper, and hope that it makes these aspects of our paper clearer.

---

> > ### Comment · Reviewer_rKBg · 2021-11-24
> > **Doubts on transferability**
> >
> > Thank you for addressing my comments. The point raised on limited morphological differences is quite important, and also relate to the question of transferability to real robots. I am not fully convinced by the argument that base position, orientation of arms, or their relative configurations are in fact morphological differences. The same holds regarding the argument that real robot experiments would only result in marginally more noisy trajectories, as it was generally observed that transferring skills in this domain can prove very challenging. Did you actually observe this by testing your method on real robots? If so, can you tell whether the transfer success is due to the spatial invariance afforded by your method rather than to the fact that the skills considered do not need high precision to be achieved? Due to the uncertainty on these points I have slightly lowered my score to 6.

---

> > > ### Author Response · Authors · 2021-11-26
> > > **Response to transferability concerns on real robots and morphologies - Part 1**
> > >
> > > We thank the reviewer for going through our responses, and for giving us additional feedback. We address their concerns on the demonstrated morphological differences, as well as applicability to real world robots below.
> > >
> > > 1. On morphological differences between demonstrated robots -
> > >     1. We would like to first clarify that our work targets a subset of the various morphological differences present between various robots. Despite considering this subset of morphological differences, we believe the correspondences our approach learns on this subset of domain pairs is tremendously powerful, and equips various robots with novel task strategies via this transfer.
> > >     2. We believe that even individual differences in robot morphologies can have a significant effect on how a robot executes a set of skills. For example, a robot with a left handed elbow joint would reach to an object to its right by bending its elbow joint, while a robot with a right handed elbow joint may need to rotate its shoulder and wrist joints to reach an object towards its right. This is an example of how individual differences, such as differences in relative orientations of joints can have a large effect on how robots perform skills; they are hence significant morphological differences. That said, we accept that certain morphological differences cited by the reviewer (such as the base position, or orientation of the arms), are not salient morphological differences on their own. However, we believe that considering these differences in conjunction with other morphological differences (link lengths, arrangement of joints, joint limits, other dynamic properties, etc.) amount to significant morphological differences between domains. Just as relative orientations have a significant effect on how robots execute skills, so too do link lengths, joint limits, and other dynamic properties such as inertia or joint velocity limits. It is owing to these differences, and their effects on how robots execute skills, that we consider the robot domains we show results on to be significantly morphologically different domains. The Baxter and Sawyer robots are particularly different in this regard.
> > >     3. Now consider the pair of domains of the Baxter Left and Right hands. Despite these robots being morphologically similar (barring their relative configurations of joints and orientations of arms), transferring skills across these robots is not trivial. For example, the state imitation baseline we consider exemplifies how traditional methods of skill transfer are likely to fail. Moreover, in order to successfully transfer skills using such methods, a large amount of manual specification of the morphological differences between domains is required. Our approach is unaware of the nature and extent of these morphological differences, but is able to transfer skills across these morphologies nonetheless, and bypass the engineering usually required to do so. The value of our experimental results is best observed in context of this difficulty of transferring skills across even “similar” morphological pairs such as the Baxter Left hand and Baxter right hand.
> > >     4. As mentioned in our responses to other reviewers, we believe our approach would facilitate transfer across the following different morphological pairs, since the fundamental premise on which our approach operates is valid for the following morphological pairs. We did not show results on these exemplar pairs owing to availability of large scale demonstration datasets on these domains.
> > >         1. Consider the example of two robot manipulators placing an object at a particular location, one equipped with a mechanical gripper, and the other equipped with a suction cup “gripper”. Irrespective of gripper morphology, both robots would follow the same strategy of first reaching towards the object, “gripping” the object (by pinching it with the mechanical gripper, or by sucking on to it), lifting the object up, and then finally placing it at the particular location and releasing its grasp. We see that our fundamental premise holds; we believe therefore that our approach would likely perform well in transferring across this morphology pair.
> > >         2. Consider the example of a wheeled robot and a legged robot navigating through a maze, each equipped with skills of moving in each of the 4 cardinal directions for a distance of 2 meters. Both of these robots would likely follow the same waypoints to navigate the maze, and hence use the same sequence of locomotion skills or strategy to navigate the maze. This implies similar skills would be observed in similar contexts, allowing our approach to successfully exploit this information to successfully transfer skill across these morphologies. We note that other works also use similar ideas; notably Hejna et. al. (2020) address morphological transfer in robot locomotion using a similar idea of high-level strategies in locomotion being shared across robots.

---

> > > ### Author Response · Authors · 2021-11-26
> > > **Response to transferability concerns on real robots and morphologies - Part 2**
> > >
> > > 1. We first continue our response to concerns on morphological differences.
> > >
> > >     5. We would like to emphasize the potential impact of our work in a broader context. We note that while a large part of the discussion on our work surrounds robot morphologies, our work provides machinery to learn unsupervised correspondences across robot domains more generally. In particular, our work serves to learn correspondences across domains such as robot environments (such as robots in either a kitchen or at a study desk), as well as various skill representations.
> > >         1. We present an additional experimental result to exemplify this potential. We trained our approach to translate between a skill space learnt in joint angle space of the Baxter Left Hand (just as presented in the paper), and a skill space learnt in end-effector space of the Baxter Left Hand. Each of these skill spaces are learnt from a mutually exclusive set of trajectories on the Baxter Left Hand. The joint space and end-effector space skill representations are significantly different representations, since they are optimized to encode overall joint positions of the arm, and the position (and orientation) of the end effector respectively. Despite this difference in representation, our approach is able to learn correspondences across these skill spaces that are consistent with the known solution of these correspondences, i.e. the forward / inverse kinematics. We observe similar trends on this domain pair as observed in Table 1, where our approach outperforms the domain adversarial training and cycle GAN baselines on the unsupervised alignment metrics, and the supervised Z error and label accuracies.
> > >
> > > 2. On the applicability of our method on real robots -
> > >     1. We have not yet evaluated our method on real world robots - unfortunately we do not currently have access to real robots. We hope to include real world robot experiments to our work in the future.
> > >     2. The reviewer raises the point of (anticipated) origin of transfer success of our pipeline on real robots. We clarify a small misunderstanding here - the spatial invariances in skills are a result of the original skill learning approach of Shankar & Gupta (2020), and not our method. Since our work learns to translate between skills from each of the source and target domains, rather than generating translated target domain trajectories directly, our work benefits from the spatial invariance of the underlying skill spaces. In particular, how well the translated skills from our approach perform on a real robot is determined by the following two factors -
> > >         1. The quality of the learnt correspondences - i.e. how well skills are translated across robots.
> > > Our paper assesses this in simulation.
> > >         2. How faithfully the skills from the original skill spaces can be executed on a real robot, or how well the skills from Shankar & Gupta (2020) would be executed on the real robot. We now argue why we believe these skills would perform well on the real robot, and only be marginally more noisy than their simulated counterparts. This serves to argue why our approach would also function well on real robots.  While Shankar & Gupta (2020) do not present results on a real world robot, they are able to outperform a similar skill-learning work, STPG (2020), in terms of reconstructing trajectories collected from real world robots. Further, STPG (2020) present visualizations of their skills being executed on a real world robot notably smoothly and without large noise. Since Shankar & Gupta (2020) are able to reconstruct trajectories more faithfully and accurately than STPG (2020), we believe executing the skills from Shankar & Gupta (2020) on a real robot would be smooth, and faithful and only marginally more noisy than their simulated versions. As a result, we believe executing our approach and overall pipeline on a real world robot would perform similarly to its performance in simulation, albeit with trajectories being marginally more noisy, as mentioned before.
> > >
> > > We hope that these clarifications serve to allay the reviewers’ concerns - firstly that the results we present are significantly morphologically different to show the efficacy of our approach, and conveying the reasons for our belief in how well our approach would work on real robots,  considering the lack of access to a real robot. We hope this removes the uncertainty the reviewer had on these points, and restores the reviewers’ confidence in our work.

---

### Official Review · Reviewer_dwTB · 2021-11-03

**Correctness:** 2
**Technical Novelty And Significance:** 2
**Empirical Novelty And Significance:** 3
**Recommendation:** 3
**Confidence:** 4

**Main Review:**


Strengths:

+ The idea of transferring skills between robots with different morphologies is interesting.

+ Presentation of the technical approach is clear.

Weaknesses:

- What are the theoretical novelties? The novelty of the paper is not clearly discussed and justified in the paper. The idea of using forward and backward losses in the objective function is widely used to maintain consistency in domain adaptation and transfer learning.

- The insight/premise that differently morphological robots follow similar strategies to address similar tasks seems questionable. For robot manipulation, this can be true. But for robot locomotion as an example, this assumption may not hold: robots can perform locomotion by walking, crawling, hopping, and wheeling, based on the morphology of the robot. How does the premise hold in general robotics tasks?

- The terms (skill, strategy, and mode) are confusing, and it is hard to imagine these terms in the context of a robotics application. Can the authors provide a concrete scenario of robots using a sequence of skills to complete a task? What are the examples of skills, strategies, and modes in that concrete scenario?

- What is the runtime (or computational cost) of the proposed approach and other compared methods? How many iterations are used for the algorithm to converge? The paper argues efficiency several times, but does not provide clear theoretical analysis or experimental support.

- How to segment a sequence of observations into skill tuples? How the start and end points of each skill are determined when segmenting a long sequence?

- “Analysing qualitative skill translations” in the experimental section 4.1 has no experimental support. No experimental results are provided to justify the statements in this subsection.

- It is not convincing to argue the left arm and the right arm of a Baxter robot have a different morphology in the experiments. The Sawyer robot and the Baxter robot’s arms also have similar design and configurations. In the application of robot manipulation, suction cups, soft grippers and traditional mechanical grippers can be considered as morphologically different robots, or maybe robot arms with obvious differences, e.g., in terms of degrees of freedom. How the proposed approach can be generalized to transfer skills among these robots with obvious morphological differences?


**Summary Of The Paper:**

This paper presents an unsupervised learning method to identify skill correspondences for translating skills between robots with different morphology. The approach is inspired by unsupervised machine translation, and uses an unsupervised objective to learn a skill translation model to match the distributions across domains.

**Summary Of The Review:**

While the idea of transferring skills between robots with different morphology is interesting, the paper is preliminary and will need improvements on explaining its novelty, premise, and terminology; more convincing experiments using robots with obvious morphological differences are also expected.

---

> ### Author Response · Authors · 2021-11-16
> **Response to Reviewer dwTB - Part 1**
>
> We would like to thank the reviewer for their detailed review and feedback on our paper. Before we address the reviewer’s concerns, we would first like to highlight our contributions and re-emphasize their novelty and significance.
>
> To the best of our knowledge, our work is the first to tackle the problem of learning unsupervised skill correspondences across robots. We believe this is a very impactful problem that greatly benefits the robot-learning community, and hope our work serves as a good starting point for future research in this direction.
>
> In order to address this problem, our work makes the following contributions -
> 1. Ideological contributions -
>     1. We note that different robots use similar strategies to solve similar tasks, and make the insight that this idea can be leveraged to learn unsupervised skill correspondences across robots - an idea that has not been explored before to the best of our knowledge.
>     2. We further make the insight that the problem of learning unsupervised skill correspondences from unlabelled demonstrations closely mirrors unsupervised machine translation (and the related field of unsupervised image translation), unlocking much of the machinery used in those communities to be used in the robotics domain.
>
> 2. In order to put these ideas to use, we introduce the following technical contributions -
>     1. We introduce an unsupervised objective to learn skill correspondences by matching distributions of sequences of skills across robots, thus operationalizing the above insights.
>     2. While ideas of combining backward and forward losses have been used before in domain adaptation, simply applying these ideas to our problem domain does not work (as noted in Section A.3). We bypass these issues by making the following small but significant contributions.
>     3. In order to prevent learning of incorrect correspondences due to ambiguity, we propose the use of a tractable form of sequential information (i.e., skill-tuples) to guide learning to semantically meaningful correspondences.
>     4. We approach the problem of matching skill sequence distributions from an explicit distribution perspective, in contrast with prior works (Zhou et. al. (2019), Ganin et. al. (2016)). To do so, we constructing explicit estimates of distributions of sequences of skills in both source and target domains, allowing for a simple maximum likelihood style objective for correspondence learning.
>
> 3. To demonstrate how well we can translate robot skills, we make the following experimental contributions -
>     1. We first show that our proposed approach can successfully learn unsupervised skill correspondences that are semantically meaningful, across various morphologically different robot pairs.
>     2. We show our approach allows for the transfer of task strategies across morphologically different robots, accelerating downstream task learning, and translation of trajectories across robots.
>
> Our work brings together ideas of learning unsupervised skill correspondences, matching skill sequence distributions, and applying them to solving robotic tasks in a novel manner; we hope the reviewer agrees that these are novel and significant contributions to an impactful problem.
>
> Having clarified the novelty of our approach, we address the specific concerns raised by the reviewer below -
> 1. On the validity of our premise across general robotic tasks - We believe our premise holds true across a wide variety of robotic applications. We discuss a few of these applications here, and hope to show by example that this premise is indeed  justified.
>     1. Consider the example of two robot manipulators placing an object at a particular location, one equipped with a mechanical gripper, and the other equipped with a suction cup “gripper”. Irrespective of gripper morphology, both robots would followthe same strategy of first reaching towards the object, “gripping” the object (by pinching it with the mechanical gripper, or by sucking on to it), lifting the object up, and then finally placing it at the particular location and releasing its grasp. We see our premise still holds in this example, enabling successful transfer of skills across these robots.
>     2. Consider the example of a wheeled robot and a legged robot navigating through a maze, each equipped with skills of moving in each of the 4 cardinal directions for a distance of 2 meters. Both of these robots would likely follow the same waypoints to navigate the maze, and hence use the same sequence of locomotion skills or strategy to navigate the maze. This implies similar skills would be observed in similar contexts, allowing our approach to successfully exploit this information to successfully transfer skill across these morphologies. We note that other works also use similar ideas; notably Hejna et. al. (2020) address morphological transfer in robot locomotion using a similar idea of high-level strategies in locomotion being shared across robots.

---

> ### Author Response · Authors · 2021-11-16
> **Response to Reviewer dwTB - Part 2**
>
> 1. On the validity of our premise across general robotic tasks (continued) -
>     3. Despite our premise holding across a wide variety of robotic applications, it would likely not hold for drastically different morphological robots. For example, consider a serial robot manipulator and a gantry robot addressing an object placing task in the presence of obstacles. The gantry robot may be able to reach positions in the workspace that a serial manipulator cannot. This enables the gantry robot to simply grasp and place the desired object, while the serial manipulator may need to rearrange the obstacles to reach the desired object at all. Such widely different morphologies may necessitate the use of widely differing strategies across robots, invaliding our premise. We would like to emphasize that these are extreme cases under which our premise does not hold true. We believe our premise is still valid in a wide variety of applications, and could facilitate knowledge transfer across many morphologies that would help accelerate robot learning.
>
>
> 2. On the terminology of skills, strategies and modes in robotics applications -
>     1. Consider the example of a Sawyer robot performing the task of stacking a block on top of another block. To solve this task, the robot would need to be equipped with “skills” such as reaching, grasping, lifting, and placing.
>     2. The “strategy” that the robot would use to solve this task can be specified by the following sequence of skills - reaching toward the block, grasping the block, lifting the block, placing the block (at a location on top of the other block). This sequence of four skills specifies the “strategy” to solve the task. These examples of skills and strategies are consistent with their definitions in robotics literature Liutikov et. al., (2013), Shankar & Gupta, (2020).
>     3. Our paper refers to “modes” of skills, and “modes” of skill-tuples. We use the term “modes” in a similar context to machine learning literature Li & Malik, (2018). Here, “modes” of skills simply refer to the various types of skills that occur in the individual skill spaces in each domain; for example, some observed modes of skills include “placing to the left”, “reaching”, “twisting”, etc. One may think of modes of skills as the various clusters of skills that emerge in the latent space of skills, as visualized in the individual skill spaces in Figure 2 of our paper. One mode of skills allows for small variations in exact joint angles, velocities that occur the skills, but largely elicits consistent behavior for all skills in that mode. Modes of skill-tuples are similarly the various skill-tuples observed in each domain. Some observed modes of skill-tuples include “reaching and grasping”, and “placing and returning”, etc.
>
> 3. On the runtime of the proposed approach versus the baselines - In addition to the brief discussion in the end of section 4.2 on number of training iterations required by each approach, we discuss the efficiency of our approach along the following two axes -
>     1. Number of training iterations and runtime of the various alignment approaches used in Table 1 - i.e., ours, Domain Adversarial training (adapted from Ganin et. al. (2016)), and CycleGAN (adapted from Zhu et. al. (2017)). Note that the State Imitation baseline does not require any training. Our approach requires roughly ~ 40-50k training iterations to converge to a good translation model. This corresponds to a runtime of ~ 3 hours. In contrast, both the domain adversarial training and Cycle GAN approaches require on the order of ~ 300-400k training iterations for their adversarial training to converge, which corresponds to a runtime of ~20-24 hours of train time. Since our approach avoids the alternating adversarial training, this drastically helps our approach converge more quickly, to translation models that also perform better. We will include this to our paper as further empirical evidence to the efficiency of our approach.
>     2. Number of RL training episodes required for the various transfer approaches used in Table 2, i.e., the hierarchical RL baseline, our approach with and without finetuning. We would like to reiterate the statistics of training episodes provided in the paper in section 4.2 for clarity. Without fine-tuning, our approach requires 0 RL episodes. With fine-tuning, our approach is trained for 50 RL episodes, as mentioned in section 4.2. This corresponds to roughly 30 minutes of train time. We observe no increase in performance upon further training. The hierarchical RL baseline (adapted from Kulkarni et. al. 2016), requires ~500 RL episodes to converge, which corresponds to about ~5 hours of training time. We hope that these two sets of experimental evidence serve to allay the reviewer’s concerns on runtime of our approach.

---

> ### Author Response · Authors · 2021-11-16
> **Response to Reviewer dwTB - Part 3**
>
> 4. On the segmentation of trajectories into skills -
>     1. We follow the method of segmentation specified by Shankar & Gupta (2020), described here. When provided with an input trajectory, the variational encoder predicts a binary variable at every timestep, that determines whether to continue executing a particular skill, or to begin executing a new skill, i.e. the skill segmentation.
>     2. We construct skill tuples by simply treating consecutive pairs of segmented skills as a single skill tuple, as mentioned in section 3.3.
>
> 5. On the presentation of experimental results for section 4.1 -
>     1. We believe the extensive qualitative results presented in our project webpage https://sites.google.com/view/translatingrobotskills/home, as mentioned in the beginning of section 4.1, along with Figure 1, do indeed support our claims. Specifically, the “Translating Individual Skills” section of the webpage presents results that show that the learnt correspondences are semantically meaningful at a coarse level (for example, placing corresponds with placing, reaching with reaching, etc.), but are unable to capture finer correspondences (such as the shape of arm across reaching skills), across all 3 domain pairs. Figure 1 shows a specific instance of this across all 3 domain pairs, where the arm shape of reaching motions does match, but the semantics of reaching skills are captured well. Since these qualitative results are best visualized dynamically (as GIFs), and owing to space constraints, we were unable to include these extensive visualizations in our main paper, besides Figure 1.
>
> 6. On the morphological differences of the presented domains -
>     1. While we agree the morphological differences cited by the reviewer (different numbers of joints, etc.) are more salient morphological differences, we believe the robot domains we consider are indeed significantly different. For example, the Baxter and the Sawyer differ in terms of the base position and orientation of their arms, their relative configurations, link lengths, arrangement of joints, joint limits, and their kinematic and dynamic properties. The Baxter left and right hands differ in their relative orientations of joints with respect to one another. Transferring skills across these morphologies traditionally requires a large amount of engineering effort to specify the nature of these morphological differences. Even with such engineering, such approaches are not guaranteed to be successful, as exemplified by our State Imitation baseline. Our approach is unaware of the nature of these morphological differences, but is able to transfer skills across these morphologies nonetheless, and bypasses the heavy robotic engineering usually required to do so. We hope this convinces the reviewer of the value of the experimental results we present.
>     2. As we mention in Section 4, our choices of robots are restricted by the availability of demonstration datasets collected through kinesthetic teaching (Sharma et. al. 2018, on the Baxter) or tele-operation (Mandlekar et. al. 2018, on the Sawyer). We choose to use these datasets rather than artificially generated data, to ensure our approach is able to learn correspondences on real-world tasks, and does not exploit artifacts of synthetic data in order to learn correspondences.
>     3. Given these practical constraints, we believe the robot domain pairs we show results on serve to conclusively showcase the efficacy of our approach. To further demonstrate the potential of our approach to transfer skills across broader morphological differences, we plan to include results on other robot domain pairs, including versions of the Sawyer and the Baxter that have one or more of their joints frozen to a particular position, or the Franka robot as used in Gupta et. al. (2019). Collecting the required data and carrying out these experiments within the rebuttal timeframe would be difficult; we propose to add them to the camera ready version of our paper if accepted.
>     4. We believe our approach would be able to effectively transfer skills across more general morphological differences (such as across the Franka, or the crippled Sawyer / Baxter).
>         1. As mentioned in Sections 4.1 and A.2, our method translates skills that occur in similar contexts across robot domains, rather than specification of the nature of the morphological differences. As a result, the extent of the morphological differences between the two robot domains is immaterial, so long as the robots exhibit skills in similar contexts in the observed demonstration data, or exhibit similar strategies to solve similar tasks.
>         2. The examples cited above (in points 1 (a), 1 (b), and 1 (c) ) regarding the validity of our premise also show how well our approach would function in translating robot skills across robots with more obvious morphological differences. We direct the reader to these examples again, and omit them here for brevity.

---

> ### Author Response · Authors · 2021-11-16
> **Response to Reviewer dwTB - Part 4**
>
> Summary -
>
> We hope that our clarifications serve to address the concerns raised by the reviewer, and help allay their concerns of explaining the novelty, premise and terminology of our work.  We shall add these clarifications to our paper, and hope that it makes these aspects of our paper clearer. We hope the clarifications on our experiments help convince the reviewer of the validity and significance of our experiments, and help them appreciate the potential impact of our work.
>
>
> References -
>
> Liutikov et. al. (2016) - Learning Manipulation by Sequencing Motor Primitives with a Two-Armed Robot, Rudolf Lioutikov, Oliver Kroemer, Guilherme Maeda and Jan Peters, Intelligent Autonomous Systems 13. Springer, 2016, pp. 1601–1611.
> Gupta et. al. (2019) - Relay Policy Learning: Solving Long-Horizon Tasks via Imitation and Reinforcement Learning, Abhishek Gupta, Vikash Kumar, Corey Lynch, Sergey Levine, Karol Hausman, arXiv preprint arXiv:1910.11956, 2019.
> Zhou et. al. (2019) - Chunting Zhou, Xuezhe Ma, Di Wang, and Graham Neubig. Density matching for bilingual word embedding. In Meeting of the North American Chapter of the Association for Computational Linguistics (NAACL), Minneapolis, USA, June 2019. URL https://arxiv.org/abs/ 1904.02343.
>
> Shankar & Gupta (2020) - Tanmay Shankar and Abhinav Gupta. Learning robot skills with temporal variational inference. In Proceedings of the 37th International Conference on Machine Learning, volume 119 of Proceedings of Machine Learning Research, pp. 8624–8633. PMLR, 13–18 Jul 2020. URL https://proceedings.mlr.press/v119/shankar20b.html.
>
> Sharma et. al. (2018) - Pratyusha Sharma, Lekha Mohan, Lerrel Pinto, and Abhinav Gupta. Multiple interactions made easy (mime): Large scale demonstrations data for imitation. In CoRL, 2018
>
> Mandlekar et. al. (2018) - Ajay Mandlekar, Yuke Zhu, Animesh Garg, Jonathan Booher, Max Spero, Albert Tung, Julian Gao, John Emmons, Anchit Gupta, Emre Orbay, Silvio Savarese, and Li Fei-Fei. Roboturk: A crowdsourcing platform for robotic skill learning through imitation. In Conference on Robot Learning, 2018
> Li & Malik (2019) - Ke Li and Jitendra Malik. Implicit maximum likelihood estimation, 2019. URL https:// openreview.net/forum?id=rygunsAqYQ.
>
> Zhu et. al. (2017) - Jun-Yan Zhu, Taesung Park, Phillip Isola, and Alexei A Efros. Unpaired image-to-image translation using cycle-consistent adversarial networks. In Computer Vision (ICCV), 2017 IEEE International Conference on, 2017.
>
> Ganin et. al. (2016) - Yaroslav Ganin, Evgeniya Ustinova, Hana Ajakan, Pascal Germain, Hugo Larochelle, Franc¸ois Laviolette, Mario March, and Victor Lempitsky. Domain-adversarial training of neural networks. Journal of Machine Learning Research, 17(59):1–35, 2016. URL http://jmlr.org/papers/v17/ 15-239.html
>
> Kulkarni et. al. (2016) - Tejas D. Kulkarni, Karthik R. Narasimhan, Ardavan Saeedi, and Joshua B. Tenenbaum. Hierarchical deep reinforcement learning: Integrating temporal abstraction and intrinsic motivation. In Proceedings of the 30th International Conference on Neural Information Processing Systems, NIPS’16, pp. 3682–3690, Red Hook, NY, USA, 2016. Curran Associates Inc. ISBN 9781510838819
>
> Hejna et. al. (2020) - Donald Hejna, Lerrel Pinto, and Pieter Abbeel. Hierarchically decoupled imitation for morphological transfer. pp. 4159–4171, 2020.

---

### Official Review · Reviewer_rjam · 2021-11-03

**Correctness:** 3
**Technical Novelty And Significance:** 2
**Empirical Novelty And Significance:** 2
**Recommendation:** 5
**Confidence:** 3

**Main Review:**


The idea of being able to reproduce skill sequences between different robots based on videos or visual demonstrations is a very hard problem with potentially very impactful applications!

In this current work, I am not quite sure what is considered as morphologically different? Does it refer to different kinematics or robots with different DOFs?
Depending on this definition, the assumption: “different morphological robots use similar task strategies (in terms of sequences of skills) to solve similar tasks” might not always hold. Although the high-level of semantic skill sequences might be the same, the actual trajectories corresponding to those skills can be significantly different.

It would be useful to clearly emphasise the contributions of the approach, as many parts rely on TVI introduced in Shankar & Gupta (2020).

I am not sure how Eysenbach et al. (2019) and Sharma et al. (2020) are related to the presented approach. However, I think there are several older related works, which would help provide context and give motivation to the proposed approach, such as:
- Movement primitives:
  - Kober, J. and Peters, J. Learning motor primitives for robotics. ICRA, 2009.
  - Ijspeert, A. J., Nakanishi, J., Hoffmann, H., Pastor, P., and Schaal, S. Dynamical movement primitives: learning attractor models for motor behaviors. Neural computation, 2013.
- Movement scaling/retargeting:
  - https://mediatum.ub.tum.de/doc/1192176/file.pdf
  - https://hal.archives-ouvertes.fr/hal-02398106/document
  - https://hal.archives-ouvertes.fr/file/index/docid/1054887/filename/Sakka2014.pdf
  - https://www.worldscientific.com/doi/abs/10.1142/9789814623353_0008

As well as work on imitation learning, learning from demonstrations and hierarchical RL, similarly as it is discussed in the related work of Shankar & Gupta (2020).

It would also be beneficial to show the performance of the proposed approach in comparison to simple skill transfer or learning from demonstration approaches as a baseline.

Text around the images in Fig2 is too small and difficult to interpret, maybe some alternative way of displaying it would aid visibility


**Summary Of The Paper:**

This paper addresses the problem of transferring skills between morphologically different robots. This approach to learning skill correspondences is framed as a problem of matching distributions of sequences of skills across robots. The paper proposes an unsupervised objective, inspired by work in unsupervised machine translation, that makes the skill translation model learn to match the distribution of skill sequences. The proposed approach is experimentally evaluated on 3 transfer settings in a simulated robot environment.

**Summary Of The Review:**

- Clarify contributions and assumptions
- More discussion on the limits of the approach w.r.t. robot morphologies.
- Update related work discussion

---

> ### Author Response · Authors · 2021-11-17
> **Response to Reviewer rjam - Part 1**
>
> We thank the reviewer for their review and feedback on our work. We are glad they appreciate the potential impact and difficulty of our problem. As done in our response to reviewer dwTB, we would like to highlight our contributions and re-emphasize their novelty and significance.
>
> To the best of our knowledge, our work is the first to tackle the problem of learning unsupervised skill correspondences across robots. We believe this is a very impactful problem that greatly benefits the robot-learning community, and hope our work serves as a good starting point for future research in this direction. We make use of the skill learning framework from Shankar & Gupta (2020) to provide us with the initial skill spaces in each domain that we seek to translate between. Our work makes the following contributions that build on the work from Shankar & Gupta (2020):
> 1. Ideological contributions -
>     1. We note that different robots use similar strategies to solve similar tasks, and make the insight that this idea can be leveraged to learn unsupervised skill correspondences across robots - an idea that has not been explored before to the best of our knowledge.
>     2. We further make the insight that the problem of learning unsupervised skill correspondences from unlabelled demonstrations closely mirrors unsupervised machine translation (and the related field of unsupervised image translation), unlocking much of the machinery used in those communities to be used in the robotics domain.
> 2. In order to put these ideas to use, we introduce the following technical contributions -
>     1. We introduce an unsupervised objective to learn skill correspondences by matching distributions of sequences of skills across robots, thus operationalizing the above insights.
>     2. While ideas of combining backward and forward losses have been used before in domain adaptation, simply applying these ideas to our problem domain does not work (as noted in Section A.3). We bypass these issues by making the following small but significant contributions.
>     3. In order to prevent learning of incorrect correspondences due to ambiguity, we propose the use of a tractable form of sequential information (i.e., skill-tuples) to guide learning to semantically meaningful correspondences.
>     4. We approach the problem of matching skill sequence distributions from an explicit distribution perspective, in contrast with prior works (Zhou et. al. (2019), Ganin et. al. (2016)). To do so, we constructing explicit estimates of distributions of sequences of skills in both source and target domains, allowing for a simple maximum likelihood style objective for correspondence learning.
>
> 3. To demonstrate how well we can translate robot skills, we make the following experimental contributions -
>     1. We first show that our proposed approach can successfully learn unsupervised skill correspondences that are semantically meaningful, across various morphologically different robot pairs.
>     2. We show our approach allows for the transfer of task strategies across morphologically different robots, accelerating downstream task learning, and translation of trajectories across robots.
>
> Our work brings together ideas of learning unsupervised skill correspondences, matching skill sequence distributions, and applying them to solving robotic tasks in a novel manner; we hope the reviewer agrees that these are novel and significant contributions to an impactful problem.

---

> ### Author Response · Authors · 2021-11-17
> **Response to Reviewer rjam - Part 2**
>
> Having highlighted the contributions of our approach, we address the specific concerns raised by the reviewer below -
>
> 1. On morphological differences we seek to transfer across, and the validity of our premise -
>     1. While the morphological differences we demonstrate transfer across in our paper consist of robot pairs that have the same number of joints but different kinematics, we emphasize that our approach seeks to address more general morphological differences. We further explain the morphological differences our premise holds for below, and discuss the validity of our approach.
>     2. As also mentioned in our response to Reviewer dwTB, we believe the robot domains we consider are significantly morphologically different. For example, the Baxter and the Sawyer differ in terms of the base position and orientation of their arms, their relative configurations, link lengths, arrangement of joints, joint limits, and their kinematic and dynamic properties. The Baxter left and right hands differ in their relative orientations of joints with respect to one another. Transferring skills across these morphologies traditionally requires a large amount of engineering effort to specify the nature of these morphological differences. Even with such engineering, such approaches are not guaranteed to be successful, as exemplified by our State Imitation baseline. In contrast, our approach is unaware of the nature of these morphological differences, but is able to transfer skills across these morphologies nonetheless, and bypasses the heavy robotic engineering usually required to do so. We hope this convinces the reviewer of the value of the experimental results we present.
>     3. As mentioned in Sections 4.1 and A.2, our method translates skills that occur in similar contexts across robot domains, rather than specification of the nature of the morphological differences. As a result, the extent of the morphological differences between the two robot domains is immaterial, so long as the robots exhibit skills in similar contexts in the observed demonstration data, or exhibit similar strategies to solve similar tasks. As a result, we believe our premise holds true across a wide variety of robotic applications, and that our approach would be able to effectively transfer skills across more general morphological differences. We discuss a few of these applications here, and hope to demonstrate this by example.
>         1. Consider the example of two robot manipulators placing an object at a particular location, one equipped with a mechanical gripper, and the other equipped with a suction cup “gripper”. Irrespective of gripper morphology, both robots would followthe same strategy of first reaching towards the object, “gripping” the object (by pinching it with the mechanical gripper, or by sucking on to it), lifting the object up, and then finally placing it at the particular location and releasing its grasp. We see our premise still holds in this example, enabling successful transfer of skills across these robots.
>         2. Consider the example of a wheeled robot and a legged robot navigating through a maze, each equipped with skills of moving in each of the 4 cardinal directions for a distance of 2 meters. Both of these robots would likely follow the same waypoints to navigate the maze, and hence use the same sequence of locomotion skills or strategy to navigate the maze. This implies similar skills would be observed in similar contexts, allowing our approach to successfully exploit this information to successfully transfer skill across these morphologies. We note that other works also use similar ideas; notably Hejna et. al. (2020) address morphological transfer in robot locomotion using a similar idea of high-level strategies in locomotion being shared across robots.
>         3. Despite our premise holding across a wide variety of robotic applications, it would likely not hold for drastically different morphological robots. For example, consider a serial robot manipulator and a gantry robot addressing an object placing task in the presence of obstacles. The gantry robot may be able to reach positions in the workspace that a serial manipulator cannot. This enables the gantry robot to simply grasp and place the desired object, while the serial manipulator may need to rearrange the obstacles to reach the desired object at all. Such widely different morphologies may necessitate the use of widely differing strategies across robots, invaliding our premise. We would like to emphasize that these are extreme cases under which our premise does not hold true. We believe our premise is still valid in a wide variety of applications, and could facilitate knowledge transfer across many morphologies that would help accelerate robot learning.

---

> ### Author Response · Authors · 2021-11-17
> **Response to Reviewer rjam - Part 3**
>
> 2. On various related work -
>     1. As mentioned in Section 3.1 and Section 3.2, Shankar & Gupta (2020) provides us with a skill learning framework that affords us a continuous representation space of skills. We believe that other skill learning frameworks could also be used in place of Shankar & Gupta (2020). In particular, the works of Eysenbach et. al. (2019) and Sharma et. al. (2020) both learn continuous skill representation spaces. We believe that it may be possible to transfer skills learnt from these works using our approach, rather than the skills from TVI (Shankar & Gupta (2020)) alone.
>     2. We thank the reviewer for suggesting related works on movement primitives and motion retargeting.
>         1. We thank the reviewer for reminding us of these classical motor primitive works. We believe our section on “Skill Learning” literature in our related work section does provide sufficient context and motivation for our paper. That said, we shall include a small discussion on the suggested papers, illustrating the following. The dynamic motion primitives (DMP) the suggested papers propose use pre-defined primitive representations, rather than provide a learnt skill representation like those which we use in our work. DMPs represent individual trajectories or trajectory segments, and require other parameters (such as initial and final states, speed of motion, and shape parameters) to represent classes of trajectories. This instance level of trajectories does not lend itself to transfer as easily as the semantic level skill representation that we adopt in our work.
>         2. We thank the reviewer for suggesting additional motion retargeting literature. We note that we dedicate a small section of our Related Work section to motion targeting literature. Similar to the papers we refer to, the papers mentioned by the reviewers use carefully handcrafted kinematic models to transfer behaviors across morphologies. We will add these papers to our literature review.
>     3. On Imitation Learning and Hierarchical RL approaches as discussed in Shankar & Gupta (2020). - While we agree that discussing imitation learning and hierarchical RL approaches in our related work section would serve to provide some additional context in our paper, we would like to re-emphasize that the contribution of our work is an approach to learn unsupervised skill correspondences, rather than a skill learning framework in and of itself (as in Shankar & Gupta (2020)), or a hierarchical RL framework (as in Hejna et. al. (2020), or Kulkarni et. al. (2016)). To focus on our particular contribution, we chose to focus on other such correspondence learning approaches, rather than general IL or HRL techniques. We shall add a small discussion section on IL and HRL approaches akin to that in Shankar & Gupta (2020).
>
> 4. On comparisons against a skill transfer baseline or a learning from demonstration baseline -
>     1. We note that the State Imitation baseline that we present in our paper may be thought of as both a simple skill transfer baseline (i.e. by simply copying end-effector states across robots), as well as an upper bound on the performance of a learning from demonstration style approach. A learning from demonstration approach may either learn a policy to generate this trajectory, and then transferring the generated policy across domains, or transfer  individual trajectories and then learn a model to generate these translated trajectories. Since any learnt model in this pipeline would only reduce accuracy of transferring trajectories, the State Imitation baseline (which simply transfers trajectories without learning a policy) serves to upper bound performance on such learning from demonstration baselines. We hope this convinces the reviewer of the adequacy of our comparison against the State Imitation baseline, rather than other skill transfer baselines or Learning-from demonstration baselines.
>
> 5. On the presentation of text in figures -
>     1. We thank the reviewer for bringing this to our attention. We shall make this text more readable and accessible in our paper.
>
> Summary -
> We hope that our clarifications serve to address the concerns raised by the reviewer, and help allay their concerns of explaining the contributions, assumptions, limits and discussion of our work.  We shall add these clarifications to our paper, and hope that it makes these aspects of our paper clearer. We hope the clarifications on our experiments help convince the reviewer of the validity and significance of our experiments, and help them unequivocally appreciate the potential impact of our work.

---

> > ### Comment · Reviewer_rjam · 2021-12-01
> > **Thank you for your response.**
> >
> > I would like to thank the authors for taking the extra time and efforts to address my concerns, as well as those of other reviewers.
> > The concern I have is with the 1st conceptual contribution. I understand that using a gantry or parallel robot would produce a gap in the morphologies that cannot be overcome. However, the morphological, or rather kinematics, differences between Sawyer and Baxter are not appropriate to use for experiments in order to demonstrate the first contribution, in my opinion.
> >
> > One potential example would be to use a Franka Emika Panda arm and a Sawyer. They are both available in most popular simulators and could be used to perform similar tasks, while having different kinematics. Another approach would be to use a serial robot with more DOFs and maybe freezing some joints.
> >
> > The reason why I mentioned movement scaling/retargeting literature, is because these kinds of variations in morphology are considered significant in the robotics community.
> >
> > Although the contribution on learning unsupervised skill correspondences from unlabelled demonstrations seems to be useful, redesigning the experiments to demonstrate its effectiveness would greatly improve the impact that the paper would have on the community. It is for this reason that I will keep my current scores and recommendations.

---

### Official Review · Reviewer_KQQm · 2021-11-05

**Correctness:** 3
**Technical Novelty And Significance:** 3
**Empirical Novelty And Significance:** 3
**Recommendation:** 6
**Confidence:** 3

**Main Review:**

Strengths:

The paper avoids the need to manually annotate datasets with low-level correspondences between robots, which requires expertise when dealing with robots with different morphologies. Instead they provide correspondences over skills, which is much easier to specify, even by non-robotics-expert users.

Their experiments show superiority over baselines from the literature.

Weaknesses:

Although the Baxter and Sawyer have different dynamics, their kinematics are not that different. Yes, the link dimensions are different but they have the same number of degrees. I would have loved to see experiments on robots with different number of degrees of freedom as that is more challenging. The argument that the choice of robots is dictated by the availability of the datasets is kind of weak given experiments are run in simulations. It can't be hard to generate a similar dataset for other robots.

I would have loved to see an ablation study on the quality of skills across the robots. Surely, the data generation process needs to make sure there is some alignment of tasks across the robots. How much of this has an effect on the quality of the translated skills?

I would have also loved to see an analysis of the training time of the translation model in relation to the time it takes to learn the skill from scratch with the target robot.

One important aspect of transfer that is usually ignored, is the cost of transfer compared to learning from scratch. And the cost could be the time it takes to collect data to learn the translation model compared to learning the skills from scratch using the target robot. Of course, using data already existing avoids this but is still a problem for robot for which data doesn't exist. See [1] as an example, albeit in a different domain.

It's not clear if the translation model is evaluated on unseen skills or trajectories. My guess is no. How well does it generalize to unseen skills?

In Table 2, how come rewards in the source domain are higher than in the target for the HRL baseline? Especially between Bax-R and Bax-L. How different are these tasks when performed with either the right or left hands?

Moreover, why should we expect the translated trajectories obtain higher rewards in the target domain than in the source domain? Especially between left and right hands.

1. Makondo, Ndivhuwo, and Benjamin Rosman. "Towards improving incremental learning of manipulator kinematics with inter-robot knowledge transfer." 2019 Southern African Universities Power Engineering Conference/Robotics and Mechatronics/Pattern Recognition Association of South Africa (SAUPEC/RobMech/PRASA). IEEE, 2019.

**Summary Of The Paper:**

The paper proposed to learn skills correspondences between robots of different morphologies in an unsupervised way - without requiring paired data from the robots. They learn a skill translation model that maps skills from a source robot to a target robot. The translation model is learned by minimizing a loss composed of two components: (1) the likelihood of the translated skills under the target distribution estimated from target robot skills data, and (2) the likelihood of the target skills under the translated distribution estimated from the translated skills data.

The proposed approaches hinges on the assumption that robots with different morphologies follow similar high-level strategies when performing similar tasks. So, a sequence of tasks across the robots is used to learn the skills correspondences.

**Summary Of The Review:**

The paper proposes a novel idea of learning skills correspondences between robots with different morphologies in an unsupervised way, borrowing ideas from machine translation. The contributions are solid, with a few minor evaluation weaknesses as highlighted in the main review above.

---

> ### Author Response · Authors · 2021-11-18
> **Response to Reviewer KQQm - Part 1**
>
> We thank the reviewer for their review and feedback on our work. We are glad they appreciate the novelty and contributions of our work, the value of our experimental results. We address the reviewer’s specific concerns below -
>
> 1. On the morphological differences across domains and availability of data -
>     1. We agree that the morphological differences cited by the reviewer (such as different numbers of joints, etc.) are indeed more salient morphological differences, we believe the robot domains we consider are indeed significantly different. For example, the Baxter and the Sawyer differ in terms of the base position and orientation of their arms, their relative configurations, link lengths, arrangement of joints, joint limits, and their kinematic and dynamic properties. The Baxter left and right hands differ in their relative orientations of joints with respect to one another. Transferring skills across these morphologies traditionally requires a large amount of engineering effort to specify the nature of these morphological differences. Even with such engineering, such approaches are not guaranteed to be successful, as exemplified by our State Imitation baseline. In contrast, our approach is unaware of the nature of these morphological differences, but is able to transfer skills across these morphologies nonetheless, and bypasses the heavy robotic engineering usually required to do so. We hope this convinces the reviewer of the value of the experimental results we present.
>     2. As we mention in Section 4, our choices of robots / domain pairs are restricted by the availability of demonstration datasets collected through kinesthetic teaching (Sharma et. al. 2018, collected on the Baxter) or tele-operation (Mandlekar et. al. 2018, collected on the Sawyer). We emphasize that while we demonstrate results on simulated robots, our data itself is collected on real world robots, rather than being generated, or procedurally constructed in simulation. We choose to use these datasets rather than artificially generated data, to ensure our approach is able to learn correspondences on real-world tasks; this prevents the approach from exploiting artifacts of synthetic data resulting in spurious correspondences as noted in initial testing of our approach. As a result, we show results on the Baxter and Sawyer robots.  We hope this convinces the reviewer that using real-world data is an important facet of our approach, and that generating data to train our approach in a useful manner is not trivial.
>     3. These practical considerations notwithstanding, we believe the robot domain pairs we show results on serve to conclusively showcase the efficacy of our approach. To further demonstrate the potential of our approach to transfer skills across broader morphological differences, we plan to include results on other robot domain pairs, including versions of the Sawyer and the Baxter that have one or more of their joints frozen to a particular position, or the Franka robot as used in Gupta et. al. (2019). Collecting the required data and carrying out these experiments within the rebuttal timeframe would be difficult; we propose to add them to the camera ready version of our paper if accepted.
>
> 2. On studying the quality of skills across robots given the inherent alignment of tasks in the data generation process -
>     1. The reviewer raises the point of quality of translations given the inherent alignment of tasks in data. We first reiterate the point we make above, that the data we use is not generated, but rather collected - on suites of tasks presented in the MIME dataset (Sharma et. al. (2018)), and the Roboturk dataset (Mandlekar et. al. (2018)). We do not select tasks or demonstrations in any way to ensure alignment, we simply use the datasets as is (albeit while maintain an unseen set of trajectories across all tasks for evaluation). As a result, we have no control over the extent of inherent alignment present in the data that one would likely have with generated data - this makes performing an ablation study on the quality of skills with respect to the inherent alignability of the data difficult.

---

> ### Author Response · Authors · 2021-11-18
> **Response to Reviewer KQQm - Part 2**
>
> 2. On studying the quality of skills across robots given the inherent alignment of tasks in the data generation process (continued) -
>     2. As noted in Section 4 and Section A.2 of our paper, we qualitatively note that the datasets need to have an (imperfect) overlap of tasks. Ideally, there are at least a few similar tasks across both datasets that our approach can exploit to learn meaningful correspondences. We note our approach is most useful when there are tasks in the source domain but not the target domain (or vice versa). This is because translating task strategies across domains then allows solving tasks that the target domain does not have demonstrations of. We observe this is true across the dataset pairs that we use. If the datasets had larger task overlap, the translation of task strategies across domains would not be as useful, since the target domain robot “knows” how to solve enough tasks by virtue of target domain demonstrations. Conversely, if the datasets had smaller task overlap, learning meaningful skill correspondences may become slightly more difficult. We hope this addresses the reviewer’s question of the quality of skills with respect to the inherent extent of alignment.
>
> 3. On the runtime of the proposed approach versus the original skill learning pipeline - The reviewer raises an interesting and valid concern in the relative training time between the learning the translation model, versus learning the target skill space from scratch.
>     1. Our approach requires roughly ~ 40-50k training iterations to converge to a good translation model. This corresponds to a runtime of ~ 3 hours, which is significantly faster than the other alignment approaches (Domain Adversarial training and CycleGAN), as noted in our response to reviewer dwTB. Training the skill space on an individual domain takes much longer - it requires two phases of training (Shankar & Gupta (2020)), for roughly 150k iteration, which take a total of roughly 12-15 hours in order to train. We note that the training time for our alignment approach is thus not a limiting factor in the overall pipeline of transferring skills.
>
> 4. On the evaluation of our approach on unseen trajectories -
>     1. We note that all of the quantitative results presented in Table 1 are evaluated on a held out test set of trajectories, that the model does not have access to during train time. These trajectories are unseen, but are composed up of skills similar (but likely not identical) to those observed during training.
>
> 5. On the relative reward values across source and target domains -
>     1. We note that reward values are meant to be indicative of relative performance of various approaches within a given domain, rather than comparing performance across source and target domains. This is because of the following factors. The initial object states and goal states observed in the RL evaluation episodes are sampled from a preset distribution of initial states. By fixing the random seeds, we ensure they are the same across various approaches in a given domain for a given task. However, they may be different across domains, since the mean parameters of these distributions vary across domains. For example - across the left and right hand of the Baxter, the initial state distribution is set to be centered at a certain distance from the respective arm. The initial object position is sampled from this distribution, leading to different object states across domains. This leads to different reward values across source and target domains, that may be higher in the target domain than the source domain. We reiterate that the key observation here is the relative performance of our approach (with and without finetuning) over the baseline RL approach. We hope this clarifies the reviewer’s question about differences in reward values.
>
> Summary -
> We hope that our clarifications serve to address the concerns raised by the reviewer, and help allay their concerns of minor evaluation weaknesses of our work. We shall add these clarifications to our paper, and hope that it makes these aspects of our paper clearer. We hope the clarifications on our experiments help convince the reviewer of the validity and significance of our experiments, and help them unequivocally appreciate the potential impact of our work.

---

### Author Response · Authors · 2021-11-27
**On the reviewers’ concerns and the broader impact of our work**

We hope the reviewers have had sufficient time to go through our detailed responses; we hope our responses have served to allay their specific apprehensions about our work and allow them to unequivocally appreciate the contributions and potential impact of our paper.

Specifically, we hope they are able to appreciate each of the contributions reiterated in Part 1 of our response to reviewer dwTB; first and foremost of which is that our work is the first to tackle the problem of learning unsupervised skill correspondences across robots.

By providing a method to learn such skill correspondences, our work enables the transfer of task strategies across robots (and potentially other domain differences, as mentioned in our response to Reviewer rKBg’s follow up discussion). We believe this is a powerful development that greatly benefits the robot-learning community, and serves as a good starting point for future research in this direction.

We hope this note, together with our responses, gives the reviewers confidence in the novelty of our contributions and potential impact of our paper.

---

### Decision · Program_Chairs · 2022-01-20

**Decision:**

Reject

**Comment:**

The paper proposes an algorithm for unsupervised skill transfer between robots with different kinematics. Integral to the approach is the idea that while the robots differ, they may use similar strategies to perform similar tasks. Without access to paired data, the paper formulates the problem of learning correspondences between robots as one of matching skill distributions across robots. Drawing insights from work in machine translation, the paper proposes an unsupervised objective that encourages the model to learn to align the distribution over skill sequences. Experimental results demonstrate the ability to use learned skill correspondences to support transfer across different robots in different domains.

As several reviewers point out, the problem of learning to transfer skills across robots with different kinematic designs from video demonstrations raises a number of interesting challenges that are relevant to the robotics and learning communities. Among them, a fundamental contribution of the paper is the ability to learn skill correspondences in an unsupervised manner based on unlabeled demonstrations. The approach by which this is achieved (i.e., using distribution matching) is sensible and clearly described. While the reviewers agree on the significance of the research problem, they raise a few key concerns regarding the initial submission. Among them are questions about the nature and extent of the domain variations that the method can handle (e.g., between robots with different degrees-of-freedom); the significance of the contributions; and how this work is situated in the context of existing approaches to robot skill learning. Several reviewers question the definition of morphological variation and comment that these variations may violate the assumption that task strategies are similar across designs. The authors provided detailed feedback to each of the reviews, which helps to clarify several of these concerns. Unfortunately, several reviewers did not respond to multiple requests to update their reviews. The one who did decided to maintain their score.

The paper tackles an important problem in robot learning and the work has the potential to have significant impact on the way in which robots acquire new skills. The original submission together with the author responses suggest that there is are solid contributions here. The authors are strongly encouraged to revisit the discussion of the approach to more clearly convey the novelty of the approach and to consider experimental evaluations that better support these claims.